# Genome-wide approach identifies a novel gene-maternal pre-pregnancy BMI interaction on preterm birth

Xiumei Hong[1,*], Ke Hao[2,*], Hongkai Ji[3], Shouneng Peng[2], Ben Sherwood[3], Antonio Di Narzo[2], Hui-Ju Tsai[4,5], Xin Liu[4,6], Irina Burd[7], Guoying Wang[1], Yuelong Ji[1], Deanna Caruso[1], Guangyun Mao[1], Tami R. Bartell[8], Zhongyang Zhang[2], Colleen Pearson[9], Linda Heffner[10], Sandra Cerda[11], Terri H. Beaty[12], M. Daniele Fallin[13], Aviva Lee-Parritz[10], Barry Zuckerman[9], Daniel E. Weeks[14] & Xiaobin Wang[1,15]

Preterm birth (PTB) contributes significantly to infant mortality and morbidity with lifelong impact. Few robust genetic factors of PTB have been identified. Such 'missing heritability' may be partly due to gene × environment interactions (G × E), which is largely unexplored. Here we conduct genome-wide G × E analyses of PTB in 1,733 African-American women (698 mothers of PTB; 1,035 of term birth) from the Boston Birth Cohort. We show that maternal *COL24A1* variants have a significant genome-wide interaction with maternal pre-pregnancy overweight/obesity on PTB risk, with *rs11161721* ($P_{G \times E} = 1.8 \times 10^{-8}$; empirical $P_{G \times E} = 1.2 \times 10^{-8}$) as the top hit. This interaction is replicated in African-American mothers ($P_{G \times E} = 0.01$) from an independent cohort and in meta-analysis ($P_{G \times E} = 3.6 \times 10^{-9}$), but is not replicated in Caucasians. In adipose tissue, *rs11161721* is significantly associated with altered *COL24A1* expression. Our findings may provide new insight into the aetiology of PTB and improve our ability to predict and prevent PTB.

[1] Center on the Early Life Origins of Disease, Department of Population, Family and Reproductive Health, Johns Hopkins University Bloomberg School of Public Health, Baltimore, Maryland 21205, USA. [2] Department of Genetics and Genomic Sciences, Icahn Institute for Genomics and Multiscale Biology, Icahn School of Medicine at Mount Sinai, New York, New York 10029, USA. [3] Department of Biostatistics, Johns Hopkins University Bloomberg School of Public Health Baltimore, Maryland 21205, USA. [4] Department of Pediatrics, Feinberg School of Medicine, Northwestern University, Chicago, Illinois 60611, USA. [5] Division of Biostatistics and Bioinformatics, Institute of Population Health Sciences, National Health Research Institutes, Zhunan 35053, Taiwan. [6] Key Laboratory of Genomic and Precision Medicine, Beijing Institute of Genomics, Chinese Academy of Sciences, Beijing 100101, China. [7] Integrated Research Center for Fetal Medicine, Division of Maternal Fetal Medicine, Department of Gynecology and Obstetrics, Johns Hopkins University School of Medicine, Baltimore, Maryland 21287, USA. [8] Mary Ann & J. Milburn Smith Child Health Research Program, Stanley Manne Children's Research Institute, Ann & Robert H. Lurie Children's Hospital of Chicago, Chicago, Illinois 60611, USA. [9] Department of Pediatrics, Boston University School of Medicine and Boston Medical Center, Boston, Massachusetts 02118, USA. [10] Department of Obstetrics and Gynecology, Boston University School of Medicine, Boston, Massachusetts 02118, USA. [11] Department of Pathology and Laboratory Medicine, Boston Medical Center, Boston University School of Medicine, Boston, Massachusetts 02118, USA. [12] Department of Epidemiology, Johns Hopkins University Bloomberg School of Public Health, Baltimore, Maryland 21205, USA. [13] Department of Mental Health, Wendy Klag Center for Autism and Developmental Disabilities, Johns Hopkins University Bloomberg School of Public Health, Baltimore, Maryland 21205, USA. [14] Department of Human Genetics and Biostatistics, Graduate School of Public Health, University of Pittsburgh, Pittsburgh, Pennsylvania 15261, USA. [15] Division of General Pediatrics & Adolescent Medicine, Department of Pediatrics, Johns Hopkins University School of Medicine, Baltimore, Maryland 21205, USA. * These authors contributed equally to this work. Correspondence and requests for materials should be addressed to X.W. (email: xwang82@jhu.edu).

Preterm birth (PTB), a birth occurring before 37 completed weeks of gestation, affects 1 in 10 of all births in the U.S. and 1 in 8 African American (AA) births[1]. PTB infants are at a greater risk for neonatal and infant mortality, and a wide range of developmental and health problems[2,3]. The annual cost of PTB in the U.S. is at least $26.2 billion per year and climbing. The current strategies to identify and treat medical risk factors in early pregnancy have been minimally effective in reducing the PTB rate[4]. A major obstacle in doing so may lie in our incomplete understanding of its root causes and biological mechanisms.

Genetic factors are believed to play an important role in PTB, with estimated heritability varying from 13 to 40% (refs 5–7). Although both maternal and paternal genotypes contribute to the genetic makeup of the fetus, previous studies have suggested a significant maternal but not a paternal effect[8]. A number of candidate-gene studies of PTB have identified several candidate genetic variants[9–13]; however, most of these have not been successfully replicated. While several genome-wide associations studies (GWAS) of PTB have been conducted (such as those deposited in the Database of Genotypes and Phenotypes (dbGaP)), none of them have identified significant maternal genetic variants for PTB[14].

In the era of GWAS, many common traits including PTB have been haunted by the mystery of 'missing heritability'[15,16]. At least three possible explanations for 'missing heritability' have been proposed and tested: (1) genetic variants with effect sizes that are too small to have been detectable in studies to date (that is, $< 5,000$ mothers or infants in the current GWAS of PTB); (2) rare variants, which are not tagged by conventional genome-wide arrays that represent an important component of complex trait genetics[17]; and (3) gene × environment interactions (G × E)[15,16,18,19], which are typically overlooked in current genetic studies. The first possibility has been confirmed, and a meta-analysis to combine all of the available GWASs has been a popular method to identify such genetic variants. The second possibility has also been extensively explored in recent years using new technology and analytic tools, though the findings suggest that most rare variants have small effect sizes on phenotypes, and may therefore have only limited value in explaining heritability[17]. The third possibility could be a major cause of the 'missing' heritability, since G × E effects are accounted for in heritability estimations, but the genes would not be discovered without testing for interactions. Assessing G × E is complicated, primarily because statistical power is typically low when there are a large number of tests, or when the prevalence of environmental exposure is low. More importantly, some genetic studies are unable to accurately measure key environmental factors, especially when they vary with time[16]. As a result, tests for G × E interaction are usually omitted in standard GWAS.

We suspected that G × E may account for a considerable proportion of the 'missing heritability' for PTB, and thus we investigated this possibility on a genome-wide scale using extensive data from a well-established Boston Birth Cohort (BBC). Besides conducting a standard GWAS analysis, we are particularly interested in identifying common single-nucleotide polymorphisms (SNPs) that may significantly interact with maternal pre-pregnancy body mass index (BMI) on overall PTB risk, for the following reasons. First, maternal pre-pregnancy overweight/obesity (OWO) is prevalent in the BBC ($> 50\%$), which is significant from clinical and public health perspectives, and also ensures sufficient statistical power to identify a significant G × E interaction in this study. In addition, previous studies on the associations of pre-pregnancy BMI or OWO with PTB have yielded inconsistent results, including positive[20–23], null[24,25] or negative associations[26–28]. It is possible that such inconsistent findings may be in part due to maternal gene × pre-pregnancy BMI interaction[29], which is largely unexplored. Finally, maternal pre-pregnancy OWO is potentially modifiable, and findings from such studies may help women and health care providers to take active measures to ensure optimal BMI before pregnancy and optimal weight gain during pregnancy.

In this study, we identify maternal *COL24A1* gene variants (with rs11161721 as the top hit) that have a significant genome-wide interaction with maternal pre-pregnancy BMI category on overall PTB risk among AA mothers. This finding has been further replicated in AA mothers (but not in Caucasian mothers) from an independent cohort[14]. SNP rs11161721 is significantly associated with altered *COL24A1* expression in adipose tissue by querying an existing data set[30]. Our findings, if confirmed in other cohorts, may have the potential to provide novel targets for aetiology research and for the prediction and early intervention of PTB.

## Results

**Population characteristics of the study sample.** The discovery sample included 698 AA mothers of preterm babies (PTB cases) and 1,035 AA mothers of term babies (TB controls) from the BBC who passed quality control (QC) steps (Methods section). Compared to TB controls, mothers of PTB cases were older at delivery (t-test, $P = 0.02$), more likely to smoke during pregnancy (chi-square test, $P = 0.001$), be under high psychosocial stress during lifetime and/or during pregnancy (chi-square test, $P < 0.002$), have a history of illicit drug use (chi-square test, $P = 0.02$), and, as expected, have more gestational complications than controls (chi-square tests, all $P < 0.001$, Table 1). The rate of pre-pregnancy OWO was 53.4% in PTB cases and 48.1% in TB controls (chi-square test, $P = 0.18$). The population characteristics of the replication sample, including 436 AA mothers from the GWAS of Prematurity and Its Complication (the GENEVA study, dbGaP entry #phs000353.v1.p1) and 346 AA mothers from the NICHD Genomic and Proteomic Network (GPN) for Preterm Birth Research (the GPN study, dbGaP entry #phs000714.v1.p1; Methods section), is shown in Supplementary Table 1.

**Heritability estimate of PTB in the discovery sample.** The estimated heritability for overall PTB and spontaneous PTB in the BBC was 25 and 26%, respectively, which were not statistically significant (Supplementary Table 2). In comparison, the estimated heritability for pre-pregnancy BMI was substantial (50%). As a proof of concept, we estimated heritability of maternal height (37%), which was comparable to a previous finding using the same method (45%)[31,32].

**GWAS to identify susceptibility loci for PTB and replication.** In the discovery sample, we performed frequentist association tests with SNPTEST software (see Methods), and identified that rs149014416, an imputed SNP near 8p12, was genome-wide significantly associated with overall PTB ($P = 1.1 \times 10^{-8}$, Supplementary Table 3a and Supplementary Fig. 1a). Mothers carrying one copy of rs149014416-A allele were at a 2.3 (95% CI, 1.7–3.2) times higher risk of having a PTB baby (Supplementary Table 3a). A similar association was found for this SNP when spontaneous PTB was analysed as the outcome ($P = 1.7 \times 10^{-8}$). We also found that rs1558001 at 7q21–22 for spontaneous PTB and rs8029754 at 15q26 for gestational age at delivery (GA) were both genome-wide significant (Supplementary Fig. 1). SNP rs1558001 is located 7,326 bases downstream of the *hepatocyte growth factor (HGF)* gene; the rs1558001-T allele was associated with a higher risk of spontaneous PTB (odds ratio (OR), 1.7 (95%

**Table 1 | Population characteristics of 1,733 AA mothers from the BBC.**

| Variables* | Term birth controls ($n = 1,035$) | Preterm cases ($n = 698$) | P value† |
|---|---|---|---|
| Maternal age (years), mean (s.d.) | 28.3 (6.6) | 29.1 (6.9) | 0.02 |
| Pre-pregnancy BMI (kg m$^{-2}$), mean (s.d.) | 26.6 (6.6) | 26.9 (6.5) | 0.36 |
| *Maternal age (years)* | | | |
| <20 | 114 (11.0) | 71 (10.2) | 0.05 |
| 20–29.9 | 501 (48.4) | 310 (44.4) | |
| 30–34.9 | 235 (22.7) | 155 (22.2) | |
| ≥35 | 185 (17.9) | 162 (23.2) | |
| *Pre-pregnancy BMI (kg/m$^2$)* | | | |
| <18.5 | 37 (3.6) | 21 (3.0) | 0.18 |
| 18.5–24.9 | 447 (43.2) | 268 (38.4) | |
| 25.0–29.9 | 263 (25.4) | 211 (30.2) | |
| ≥30 | 235 (22.7) | 162 (23.2) | |
| Missing | 53 (5.1) | 36 (5.2) | |
| *Maternal smoking during pregnancy* | | | |
| Never | 836 (80.8) | 513 (73.5) | 0.001 |
| Quitter | 73 (7.1) | 57 (8.2) | |
| Current smoker | 99 (9.6) | 108 (15.5) | |
| Missing | 27 (2.5) | 20 (2.8) | |
| Alcohol drinking during pregnancy | 75 (7.2) | 61 (8.7) | 0.27 |
| Illicit drug use | 183 (17.7) | 163 (23.4) | 0.02 |
| Nulliparity | 438 (42.3) | 290 (41.5) | 0.45 |
| *Self-reported lifetime stress* | | | |
| Mild | 394 (38.1) | 220 (31.5) | <0.001 |
| Moderate | 522 (50.4) | 355 (50.9) | |
| High | 102 (9.9) | 115 (16.5) | |
| Unknown | 17 (1.6) | 8 (1.1) | |
| *Self-reported stress during pregnancy* | | | |
| Mild | 381 (36.8) | 215 (30.8) | 0.002 |
| Moderate | 456 (44.1) | 302 (43.3) | |
| High | 181 (17.5) | 172 (24.6) | |
| Unknown | 17 (1.6) | 9 (1.3) | |
| Hypertensive disorders during pregnancy | 75 (7.2) | 211 (30.2) | <0.001 |
| Diabetes/gestational diabetes | 70 (6.8) | 102 (14.6) | <0.001 |
| Delivery type: caesarean section | 318 (30.7) | 305 (43.7) | <0.001 |
| IUI‡ | 134 (12.9) | 185 (26.5) | <0.001 |
| Infant's gender: male | 527 (50.9) | 344 (49.3) | 0.50 |
| GA (years), mean (s.d.) | 39.6 (1.2) | 33.3 (3.6) | <0.001 |
| *PTB subgroup* | | | |
| Spontaneous PTB | — | 461 (66.0) | |
| Early PTB (<32 weeks of gestation) | — | 176 (25.2) | |
| IUI-related PTB | — | 185 (26.5) | |

BMI, Body mass index; IUI, Intra-uterine inflammation; PTB, Preterm birth; SD, Standard deviation; TB, Term birth.
*$n$ (%) are shown in the table, if not specified.
†Each variable was compared between PTB cases and TB controls, using $\chi^2$ and $t$-tests for categorical and continuous variables, respectively.
‡About 49 women with TB and 8 women with PTB had missing data on intra-uterine inflammation.

CI, 1.4–2.0); $P = 3.0 \times 10^{-8}$; Supplementary Table 3a). SNP rs8029754 is located in an intergenic region, and the rs8029754-G allele was significantly associated with a lower GA ($P = 1.9 \times 10^{-8}$; Supplementary Table 3a). Stratified analyses based on infant's gender revealed no additional significant genetic variants for PTB outcomes.

The two replication cohorts, the GENEVA and the GPN studies (Methods section), both had genotype data for rs1558001 and rs8029754. However, these two SNPs showed no associations with PTB outcomes in AA mothers from these replication cohorts, and the estimated ORs were not comparable to those in our discovery sample (Supplementary Table 3b). SNP rs149014416, which failed to be imputed in the two replication cohorts, was dropped from these analyses.

**Genome-wide search for SNP × pre-pregnancy BMI interactions.** Our G × E analyses with the conventional 1-degree of freedom (1-df) test revealed a genome-wide significant interaction between maternal rs11161721 (an intronic SNP in the *collagen, type XXIV*

*alpha 1* [*COL24A1*] gene) at 1p22 and pre-pregnancy BMI category on overall PTB risk ($P_{G \times E} = 1.8 \times 10^{-8}$, empirical $P_{G \times E} = 1.2 \times 10^{-8}$, Fig. 1a). There was no evidence of genomic inflation (Fig. 1b). In addition, SNP rs1324899 (an imputed SNP in the *COL24A1* gene), which was in moderate linkage disequilibrium (LD) with rs11161721 ($R^2 = 0.6$ based on haplotype frequencies estimated via the expectation-maximization algorithm), also yielded a significant interaction with pre-pregnancy BMI category ($P_{G \times E} = 3.2 \times 10^{-8}$, Fig. 1c). rs60891279, a missense SNP in the same gene, showed suggestive evidence of interaction with pre-pregnancy BMI category ($P_{G \times E} = 2.3 \times 10^{-7}$, Fig. 1c). Similar findings were observed when pre-pregnancy BMI was analysed as a quantitative trait ($P_{G \times E} = 4.8 \times 10^{-6}$ for rs11161721), or when pre-pregnancy BMI was classified as normal weight (pre-pregnancy BMI: 18.5–24.9 kg m$^{-2}$) versus OWO (pre-pregnancy BMI ≥ 25.0 kg m$^{-2}$; $P_{G \times E} = 5.9 \times 10^{-8}$ for rs11161721; empirical $P_{G \times E} = 5.0 \times 10^{-8}$).

In the G × E analyses we also attempted the 2-df test proposed by Kraft *et al.*[33] to search for combined signals of the SNP main effect and interactions with pre-pregnancy BMI category.

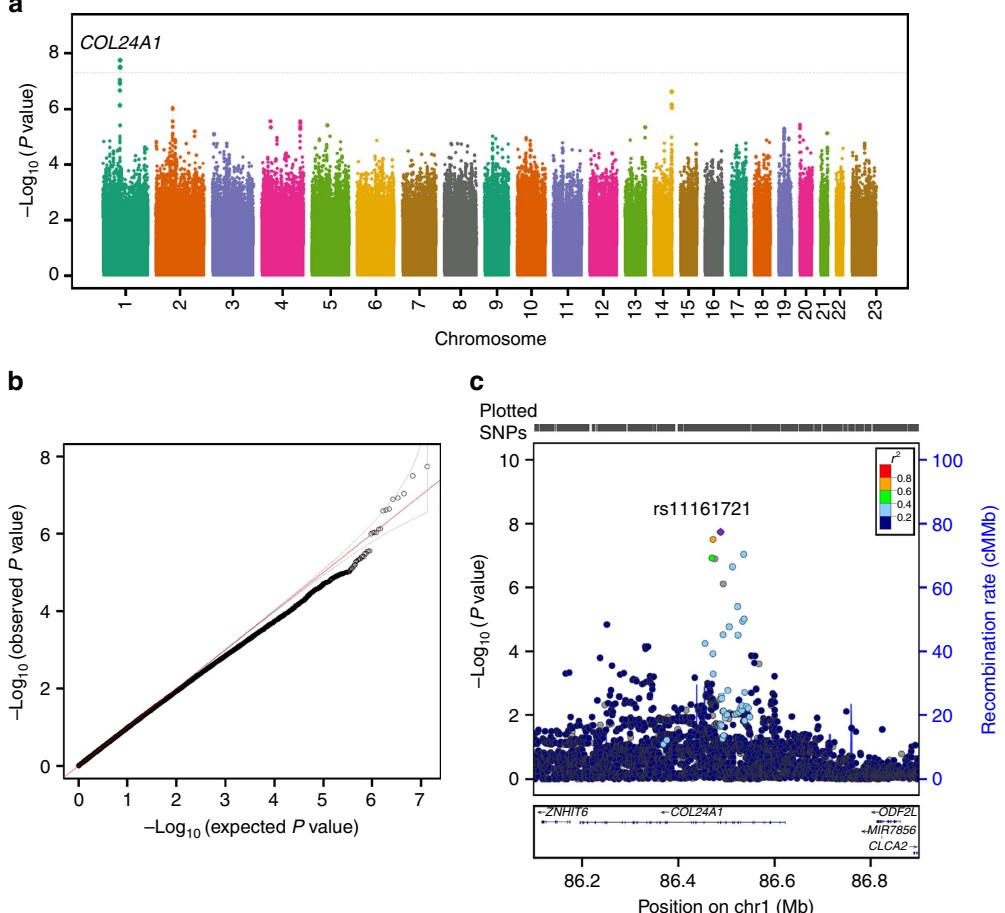

**Figure 1 | Manhattan and quantile-quantile (Q-Q) plots for genome-wide SNP interactions with pre-pregnancy BMI category on overall PTB as well as the locuszoom plot for the genomic region reaching genome-wide significance.** (**a**) Manhattan plot, (**b**) Q-Q plot for the genome-wide genotyped and/or imputed SNP interaction with pre-pregnancy BMI category and (**c**) the locuszoom plot for both genotyped and/or imputed SNPs located in the region at chromosome 1p22 that showed genome-wide significant interaction with pre-pregnancy BMI on overall PTB, in 1,586 African American mothers from the BBC. All analyses were performed using the conventional 1-degree of freedom interaction test based on the multiple logistic regression models, adjusted for genotyping batch, genetic ancestry, maternal age at delivery, parity and infant's gender.

SNP rs11161721 yielded a genome-wide borderline significant interaction with pre-pregnancy BMI category ($P_{G \times E} = 5.2 \times 10^{-8}$, Supplementary Fig. 2), but was less significant than the result of the conventional 1-df test. This may suggest that the effect of rs11161721 on overall PTB was mainly through its interaction with pre-pregnancy BMI.

***COL24A1* × pre-pregnancy BMI interaction across PTB subtypes.** The joint and interaction associations between rs11161721 and pre-pregnancy BMI category on overall PTB in the discovery sample are presented in Fig. 2. Using the normal weight mothers carrying the CC genotype as the reference group, the risk of overall PTB was 1.8 (95% CI, 1.3–2.4) and 2.0 times (95% CI, 1.4–2.8) higher in their overweight and obese counterparts, respectively; however, the risk decreased by 50% (OR, 0.5 (95% CI, 0.2–1.6)) in obese mothers carrying the AA genotype compared with normal weight mothers carrying the CC genotype. We also performed stratified analyses by maternal genotype at rs11161721, as shown in Table 2. We found that pre-pregnancy overweight and obesity significantly increased PTB risk in mothers carrying the CC genotype, but they tended to decrease PTB risk in mothers carrying the AA genotype (Table 2).

We then examined if there was heterogeneity in the magnitude and direction of the rs11161721 × pre-pregnancy BMI interaction effect across PTB subtypes. As shown in Supplementary Table 4, we found that, with the 1-df test, the interaction effect size and direction between rs11161721 and pre-pregnancy BMI category was comparable across all subtypes of PTB, including spontaneous PTB ($P_{G \times E} = 1.2 \times 10^{-5}$), medically indicated PTB ($P_{G \times E} = 1.7 \times 10^{-5}$), early PTB ($P_{G \times E} = 0.003$), late PTB ($P_{G \times E} = 1.3 \times 10^{-7}$), PTB with ($P_{G \times E} = 3.0 \times 10^{-4}$) and without intra-uterine inflammation (IUI, $P_{G \times E} = 7.4 \times 10^{-7}$), and GA as a continuous outcome ($P_{G \times E} = 3.5 \times 10^{-5}$). Further removal of mothers with obesity-related gestational complications (such as diabetes/gestational diabetes and/or hypertensive disorders) and further adjustment for other covariates did not significantly change the estimated interaction effect (Supplementary Table 5). Similar interaction associations were also observed when stratified by infant's gender (Supplementary Table 6).

**Replication of rs11161721 × pre-pregnancy BMI interaction.** Since there was no available maternal pre-pregnancy BMI information in the GENEVA study, the replication analysis for interaction between rs11161721 and pre-pregnancy BMI was conducted only in the GPN study. Due to a limited number of AA mothers in the GPN study, we coded pre-pregnancy BMI as normal weight versus OWO. This classification did not significantly change our findings in the BBC (1-df test, $P_{G \times E} = 5.9 \times 10^{-8}$;

permutation test, empirical $P_{G \times E} = 5.0 \times 10^{-8}$, Table 3). In the GPN study, a significant interaction between rs11161721 and pre-pregnancy OWO was found in 300 AA women with pre-pregnancy BMI $\geq 18.5 \, \mathrm{kg \, m^{-2}}$ (1-df test, $P_{G \times E} = 0.01$) and the association direction was consistent with that in the discovery cohort. We further tested whether the validated $G \times E$ interaction could be detected in 683 Caucasian mothers from the GPN study. However, no such interaction was observed for this SNP (Table 3) in Caucasians.

**Meta analyses on SNP × pre-pregnancy BMI interaction.** We then ran a genome-wide meta-analysis on SNP × pre-pregnancy BMI interaction for all the genotyped and/or imputed variants available in both the discovery and the replication AA women ($n = 1,886$). We found that *rs11161721* remained the top SNP yielding a genome-wide significant interaction with maternal pre-pregnancy BMI category ($P_{G \times E} = 3.6 \times 10^{-9}$ based on Stouffer's Z test for meta-analyses). SNPs rs1324899, rs60891279 and rs10443169 in the *COL24A1* gene, which were all in moderate LD with *rs11161721* ($R^2 \geq 0.4$), also met the significance threshold (Supplementary Fig. 3). No other SNPs/variants were found to have significant interactions with pre-pregnancy BMI category.

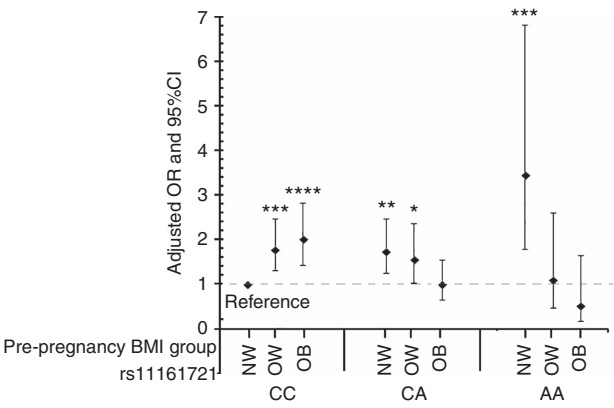

**Figure 2 | The joint associations between rs11161721 in the *COL24A1* gene and pre-pregnancy BMI on overall PTB risk.** *Y* axis reflects the OR and 95% confidence interval (CI) of overall PTB risk for each subgroup stratified by the genotype of rs11161721 and maternal pre-pregnancy BMI category, with normal weight mothers carrying the rs11161721-CC genotype as the reference group. This analysis was conducted based on the multiple logistic regression models, adjusted for genotyping batch, genetic ancestry, maternal age at delivery, parity and infant's gender. *$P < 0.05$; **$P < 0.01$; ***$P < 0.001$, ****$P < 0.0001$ when compared with the reference group. NW, normal weight; OB, obesity; OW, overweight.

**Testing for interactions in mother–infant pairs.** In a subset of infants from the BBC ($n = 153$), we found a similar but relatively weak interaction between newborn genotype at rs11161721 and maternal pre-pregnancy BMI on overall PTB risk (1-df test, OR, 0.4 (95% CI, 0.2–1.0), $P_{G \times E} = 0.05$, Supplementary Table 7). Similar findings were noticed in 276 infants from the GPN study (Supplementary Table 7).

With available GWAS data in 120 mother–infant pairs from the BBC and in 276 mother–infant pairs from the GPN study, we coded the number of maternal-origin *rs11161721-A* allele and paternal-origin *rs11161721-A* allele for each infant (Methods section), and tested their interactions with pre-pregnancy BMI category on overall PTB. Among the 120 AA infants from the BBC, the observed effect size for the maternal pre-pregnancy BMI × maternal-origin *rs11161721-A* allele interaction was higher than that for the maternal pre-pregnancy BMI × paternal-origin *rs11161721-A* allele interaction, although neither was statistically significant. In comparison, among the 276 AA infants from the GPN study, the interaction between pre-pregnancy BMI category and maternal-origin *rs11161721-A* allele was statistically significant (1-df test, $P_{G \times E} = 0.004$), while no significant interaction was found between pre-pregnancy BMI and the paternal-origin *rs11161721-A* allele, suggesting a significant maternal effect (Supplementary Table 8).

**Functional annotation of rs11161721.** Using existing expression quantitative trait loci (eQTL) data sets in adipose tissue (Methods section)[30], we found that the *rs11161721-A* allele was associated with higher *COL24A1* gene expression than the *rs11161721-C* allele in subcutaneous adipose tissue ($P = 8.2 \times 10^{-8}$) and in omental adipose tissue ($P = 2.7 \times 10^{-5}$) based the linear regression model. The association in subcutaneous adipose tissue remained significant after multiple testing corrections with a false discovery rate $< 5\%$.

**Discussion**

To our knowledge, this is the first study to date in AA women to explore genome-wide $G \times E$ interactions for PTB risk, with a particular focus on the pre-pregnancy BMI category (a common and modifiable predictor of adverse pregnancy outcomes). We identified and replicated a significant interaction between maternal genotype rs11161721 in the *COL24A1* gene and pre-pregnancy BMI category on overall PTB risk in the BBC and in an independent GWAS data set deposited in dbGaP, respectively. Although three other SNPs were found to be significantly associated with different PTB outcomes in the BBC, the associations were not confirmed in either of the two

**Table 2 | Stratified analyses\* by genotypes of rs11161721 for the association between pre-pregnancy BMI category and PTB in the mothers from the BBC.**

| Genotype | Normal weight mothers | | | Overweight mothers | | | | Obese mothers | | | | $P_{G \times E}$† |
|---|---|---|---|---|---|---|---|---|---|---|---|---|
| rs11161721 | PTB (*n*) | TB (*n*) | OR | PTB (*n*) | TB (*n*) | OR (95% CI)‡ | *P* value | PTB (*n*) | TB (*n*) | OR (95% CI)‡ | *P* value | |
| CC | 136 | 293 | 1.0 | 140 | 168 | 1.8 (1.3–2.4) | $4.3 \times 10^{-4}$ | 117 | 127 | 2.0 (1.4–2.8) | $5.4 \times 10^{-5}$ | |
| CA | 107 | 137 | 1.0 | 62 | 78 | 0.9 (0.6–1.4) | 0.60 | 41 | 93 | 0.6 (0.4–0.9) | 0.02 | |
| AA | 25 | 17 | 1.0 | 9 | 17 | 0.4 (0.1–1.2) | 0.10 | 4 | 15 | 0.2 (0.0–0.7) | 0.02 | $1.8 \times 10^{-8}$ |

CI, confidence interval; OR, odds ratio; PTB, preterm birth; TB, Term birth.
*Normal weight mothers as the reference group in each genotype strata.
†The interaction effect was analysed in the total sample by adding pre-pregnancy BMI category, *rs11161721* (under the additive genetic model) and their interaction term into the regression model, with adjustment of the same covariates as mentioned below.
‡Adjusted for genotyping batch, genetic ancestry, maternal age at delivery, parity and infant's gender.
Normal weight: pre-pregnancy BMI: 18.5–24.9 kg m$^{-2}$; overweight: pre-pregnancy BMI: 25.0–29.9 kg m$^{-2}$; and obesity: pre-pregnancy BMI $\geq 30$ kg m$^{-2}$.

**Table 3 | The main effects of maternal rs11161721-A allele, pre-pregnancy OWO and their interaction effects on PTB in the mothers from the BBC and from the GPN study.**

| Variable | BBC discovery* $n = 1,586$ | | | GPN replication[†] AA mothers $n = 300$ | | | GPN replication[†] caucasian mothers $n = 683$ | | |
|---|---|---|---|---|---|---|---|---|---|
| | OR | 95% CI | P value | OR | 95% CI | P value | OR | 95% CI | P value |
| rs11161721-A allele | 1.8 | 1.4–2.3 | $9.8 \times 10^{-6}$ | 1.4 | 0.7–2.4 | 0.32 | 0.9 | 0.7–1.2 | 0.59 |
| OWO | 1.9 | 1.5–2.5 | $2.6 \times 10^{-6}$ | 1.9 | 1.0–3.4 | 0.04 | 1.2 | 0.8–1.9 | 0.36 |
| rs11161721 × OWO interaction | 0.4 | 0.3–0.5 | $5.9 \times 10^{-8}$[‡] | 0.3 | 0.2–0.8 | 0.01 | 1.2 | 0.7–2.0 | 0.39 |

BBC, Boston Birth Cohort; CI, confidence interval; GPN, Genomic and Proteomic Network for Preterm Birth Research; OWO, overweight and/or obesity (pre-pregnancy BMI $\geq 25$ kg m$^{-2}$, compared to normal weight mothers); OR, odds ratio; PTB, preterm birth.
*The analysis was conducted using the logistic regression model, adjusted for genotyping batch, genetic ancestry, maternal age at delivery, parity and infant's gender in the BBC discovery sample.
[†]The analysis was conducted using the logistic regression model, adjusted for genetic ancestry, parity and infant's gender in the GPN replication sample.
[‡]Here P value for the interaction effect was estimated using an interaction term of pre-pregnancy OWO (coded as 0 = NW, 1 = OWO) and the rs11161721 genotype (additive genetic model), rather than an interaction term of the pre-pregnancy BMI category (coded as 0 = NW, 1 = OW and 2 = OB) and rs11161721 (additive genetic model) as shown in Table 2.

independent data sets. Taken together, these findings offer new insight into the 'missing heritability' of PTB, and highlight the importance of taking non-genetic factors into account when conducting genetic association studies of PTB.

Our findings help to explain the inconsistent findings between pre-pregnancy BMI and PTB outcomes reported in previous studies[20–28] and suggest that such inconsistency may be partly due to the interactions between maternal genotype (that is, rs11161721) and pre-pregnancy BMI. We showed that the impact of pre-pregnancy BMI on overall PTB risk depends on the maternal genotype at rs11161721. Notably, the risk of overall PTB in obese mothers carrying the CC genotype was about two times higher than in their normal weight counterparts, but the risk decreased by > 50% in obese mothers carrying the AA genotype. An alternative explanation might be that the impact of the SNP depends on maternal pre-pregnancy BMI, since in normal weight women the A-allele at rs11161721 was associated with a higher risk of having overall PTB but in obese women it was associated with a lower risk compared to the C-allele.

Our study suggests that the magnitude and direction of the rs11161721 × pre-pregnancy BMI interaction associations were comparable across PTB subtypes, including both spontaneous and medically indicated PTB, despite the fact that these two subtypes may have distinct clinical features. Consistently, a previous nationwide cohort study showed that maternal obesity could increase the risk of both medically indicated and spontaneous PTB, though via different pathways[20]. The obesity-related excess risk of medically indicated PTB may largely be due to obesity-related pregnancy disorders[20,22]. Moreover, maternal obesity has been linked to enhanced inflammation[34] and an increased rate of chorioamnionitis and/or infection[35,36], and thus it may potentially increase the risk of spontaneous PTB via inflammatory/infection up-regulation. Further studies are needed to explore how rs11161721 might be involved in these different pathways by which maternal pre-pregnancy OWO can affect both medically indicated and spontaneous PTB.

The fetal membranes are known to express one maternal allele and one paternal allele for each genetic variant. Our analyses using the AA mother–infant pairs from the replication cohort indicated that the rs11161721 × pre-pregnancy BMI interaction is due to a maternal rather than a paternal effect. Consistently, previous studies have also shown that maternal genetic factors, but not paternal genetic factors, contribute significantly to PTB risk[8].

The functionality of SNP rs11161721 in PTB remains to be investigated. It is located in the intronic region of the COL24A1 gene, and our eQTL analysis showed that the rs11161721-A allele may upregulate the COL24A1 gene expression level in adipose tissue. The protein encoded by the COL24A1 gene is involved in regulating type I collagen fibrillogenesis at specific anatomical

locations during fetal development[37]. This protein is a component of the extracellular matrix (ECM), which plays an important role in determining cell and organ function. Defects in ECM synthesis and metabolism and the physiological process of ECM turnover may contribute to changes in the fetal membranes preceding normal parturition and may influence pathological events leading to PTB[38]. However, it is unclear how collagen XXIV itself, independently or through biological interaction with maternal obesity, is actually involved in the PTB pathogenesis.

The implications of G × E interaction could be far reaching. Besides explaining some of the 'missing heritability', the discovery of significant G × E creates the opportunity for translational actions. Such findings may reveal aetiologic genes/pathways interacting with environmental factors, and identify possible drug targets. Our findings, if further validated in other independent cohorts, could also prove valuable for the prediction and prevention of PTB in AAs, since pre-pregnancy BMI is a modifiable factor and both rs11161721 genotypes and pre-pregnancy BMI can be easily measured well before pregnancy. For example, our data suggest that both normal-weight women carrying the rs11161721-AA genotype and obese women carrying the rs11161721-CC genotype are at high risk for PTB. To effectively reduce the incidence of PTB in these at-risk women, different interventions may be developed according to genetic background and pre-pregnancy BMI status: obese women carrying the rs11161721-CC genotype could reduce their PTB risk by optimizing their weight before pregnancy and/or optimizing their weight gain during pregnancy; while normal weight women carrying the rs11161721-AA genotype might protect against PTB by decreasing COL24A1 expression and/or blocking biological pathways involving collagen XXIV; however, these strategies obviously would require further investigation.

Although our standard GWAS analyses identified several genome-wide significant genotyped SNPs with marginal effects on PTB outcomes, none of them have been successfully replicated in other independent populations. It is possible that this failure to replicate may reflect differences in environmental exposures between the BBC and the available replication cohorts, given the fact that some genetic effects are dependent on them. Another possibility is that the sample sizes of the replication cohorts were too small to confirm a relatively modest effect size. A third possibility is that the associations identified in the BBC are false positives. Of note, the current study had ∼80% power to identify an OR of 2.0 or above for a SNP with MAF $\geq 10\%$ at a genome-wide significance level ($P < 5 \times 10^{-8}$), which allowed us to conclude that common genetic variants with large effect sizes, that is, OR $\geq 2$, were unlikely to substantially contribute to overall PTB risk in AA women. However, it is possible that common variants with smaller effect sizes or rare variants with larger effect sizes

(which we did not have sufficient power to interrogate in this study) remain to be identified. Meta-analysis to combine all of the available PTB GWAS should be considered in future studies.

Several limitations should be acknowledged. First, maternal pre-pregnancy BMI in the BBC was based on self-reported height and weight, thus it may be subject to some reporting bias. Nevertheless, in a subset of women, we compared self-reported BMI with that from maternal medical records and found a high degree of agreement ($r = 0.92$). Second, the replication sample from the GPN study used a more stringent case definition (all cases were those with spontaneous PTB at <34 weeks of gestation) than was used in the BBC discovery sample. We believe this difference would not introduce bias because: (1) the rs11161721-pre-pregnancy BMI interaction was comparable across different PTB subtypes including early PTB and spontaneous PTB; and (2) the use of a more stringent definition of cases in a study cohort with a limited sample size, as in our replication sample from the GPN study, may increase power by creating a relatively homogenous group of cases, and thus provide an increased opportunity to confirm a real but relatively modest interaction effect. Third, the statistical power of this study may be limited, especially for those SNPs with a low minor allele frequency (that is, MAF<10%) and/or SNPs with relatively modest interaction effects with pre-pregnancy BMI. Finally, our finding on the interaction effect between pre-pregnancy BMI and rs11161721 (as well as other SNPs in LD with rs11161721) was not observed in Caucasian mothers of the GPN study, which may have been due to the following reasons: (1) there were some residual confounding factors not considered in our analyses of Caucasian mothers; (2) the interaction effect size in Caucasians was relatively modest, and thus the current study in 683 Caucasian mothers would have limited power to identify such an interaction effect; (3) the identified interaction effect may be specific to AAs, and/or (4) we still could not rule out the possibility that our study finding in AAs is a false positive result, although such chance is extremely low, given all of the analyses and an independent replication that we performed. Further replications in larger AA and Caucasian cohorts are still needed.

In conclusion, this is the first genome-wide G × E study of PTB in AA populations. Our study identified and replicated a genome-wide significant interaction between maternal rs11161721 in the *COL24A1* gene and pre-pregnancy BMI category on overall PTB in AA mothers from the BBC and from an independent GWAS data set deposited in dbGaP, respectively. These findings highlight the importance of taking non-genetic factors into account when conducting genetic association studies of PTB. If further validated, these findings may provide new insight into the aetiology of PTB and improve our ability to predict and prevent PTB. More work remains to be done in future studies to explore interactions between genetic variants and other environmental factors.

## Methods

**The boston birth cohort (BBC).** This study used a case–control design and was comprised of the discovery and independent replication samples. The discovery sample was enroled from the BBC, a cohort initiated in 1998 with a rolling enrolment in Boston, MA, as detailed elsewhere[39]. The BBC targeted all mothers who delivered singleton live preterm or low birthweight (<2,500 g) infants, and control mothers who delivered term normal birthweight infants. Pregnancies that were a result of *in vitro* fertilization, multiple gestations (for example, twin and triplets), fetal chromosomal abnormalities or major birth defects were excluded. Each enroled participant, after giving written informed consent, completed a questionnaire interview to assess pre-pregnancy weight, height, race/ethnicity, education, smoking, parity, psychosocial stress, lifestyle and prenatal multivitamin intake. A maternal blood sample was obtained within 48–72 h after delivery and a cord blood sample was obtained at delivery. The study protocol was approved by the Institutional Review Boards (IRB) of Boston University Medical Center, the Ann & Robert H. Lurie Children's Hospital of Chicago (formerly Children's Memorial Hospital), and the Johns Hopkins Bloomberg School of Public Health.

**Study sample included in the current GWAS.** In the discovery stage, a total of 732 AA mothers of preterm babies (PTB, <37 weeks of gestation, Cases) and 1,102 AA mothers of term babies (TB, >37 weeks of gestation, Controls) were enroled from the BBC. The PTB cases and TB controls were frequency matched on race, maternal age at delivery ( ± 5 years), parity and year of delivery. The phenotypes and covariates are described below.

In the replication stage, we used two PTB GWAS data sets deposited in dbGaP to validate our GWAS findings in the BBC:GWAS of Prematurity and Its Complication (the GENEVA study, dbGaP entry #phs000353.v1.p1), and the NICHD Genomic and Proteomic Network for Preterm Birth Research (the GPN study, dbGaP entry #phs000714.v1.p1). In the GENEVA study, PTB cases were defined as mothers of babies born at 23–36 weeks of gestation, and TB controls were defined as mothers of babies born at 37–42 weeks. In the GPN study[14], PTB cases were defined as mothers of babies with spontaneous PTB at 20–33[6/7] weeks, and TB controls were defined as mothers of babies born at 39–41[6/7] weeks and were matched with cases on race/ethnicity, maternal age and parity (yes/no).

**Phenotype definition and covariates in the discovery stage.** In the BBC, GA was assessed by early (<20 weeks) prenatal ultrasound and/or based on the first day of the last menstrual period (if early prenatal ultrasound was not available) as recorded in the maternal medical record[39]. Overall PTB was defined as a birth occurring at <37 weeks of gestation. As subtypes of PTB, spontaneous PTB was defined as a birth occurring secondary to documented active preterm labour (uterine contractions with cervical effacement and dilation at <37 weeks) or premature rupture of membranes at <37 weeks without uterine contractions or both. Medically indicated PTB was defined as a birth delivered by medical induction or caesarean section at <37 weeks without uterine contractions or rupture of membranes. Early PTB was defined as a birth occurring before 32 weeks, and late PTB as a birth occurring from 32 to 36[6/7] weeks. Intra-uterine infection (IUI) of maternal origin was defined based on clinical signs of chorioamnionitis (that is, intrapartum fever >38 °C) and/or histologic chorioamnionitis[40]. PTB with IUI was defined as a birth with IUI occurring at <37 weeks.

Maternal pre-pregnancy BMI was calculated as weight (kg) divided by height squared (m²), based on self-reported pre-pregnancy height and weight, and then categorized into four groups: underweight (<18.5 kg m⁻²), normal weight (18.5–24.9 kg m⁻²), overweight (25–29.9 kg m⁻²) and obesity (≥30 kg m⁻²). The following maternal variables were defined based on a standard maternal questionnaire interview[41]: maternal smoking during pregnancy, which was classified into three groups: never smoker (did not smoke cigarettes throughout the index pregnancy), quitter (only smoked in the 3 months before pregnancy or during the first trimester) or continuous smoker (smoked continuously from pre-pregnancy to delivery); maternal psychosocial stress during lifetime and during pregnancy, which was self-reported and classified into three categories: mild, moderate and high; and maternal illicit drug use, which was self-reported and classified into no versus yes. Maternal complications during pregnancy including diabetes/gestational diabetes and hypertensive disorders were collected via maternal electronic medical record abstraction.

**Genotyping and quality control in the discovery stage.** For each participant, genomic DNA was isolated from EDTA-treated peripheral white blood cells and quantified using a Quant-iT Broad-range dsDNA Assay Kit on a SpectraMax M2 micro-plate reader. A total of 1,910 samples (including 1,834 study samples and 76 study sample duplicates) were sent to the Center for Inherited Disease Research (CIDR) for genome-wide genotyping. These maternal samples, as well as 62 HapMap duplicate samples, were genotyped in two batches: batch 1 in 2011 using the Illumina HumanOmni2.5-4v1 array, and batch 2 in 2014 using the Illumina HumanOmni2.5-8v1-1 array. Within each batch, cases and controls were balanced across 96-well plates, and each plate contained 2–4 HapMap controls and 2–4 study sample duplicates. The two batches were merged into a single data set with 2,369,543 probes in common. A total of 1,813 samples (98.9%), 76 duplicates and 62 HapMap controls were successfully genotyped and passed CIDR's QC process.

The quality assurance/QC analyses team at the University of Washington Genetics Coordinating Center (UWGCC) performed GWAS data cleaning on the combined samples according to the protocol described by Laurie *et al.*[42] using the R packages GWASTools[43] and SNPRelate[44]. Briefly, the following parameters were examined: (1) CIDR technical filters; (2) missing call rate per SNP, per chromosome and per sample; (3) the reproducibility rate among the 76 duplicated samples (24 cross-phase and 52 within-phase duplicates); (4) duplicate discordance estimates for each SNP to infer SNP quality; (5) Mendelian error check of 8 HapMap trios/duos; (6) Hardy–Weinberg equilibrium test; (7) genotyping batch effects: measured by comparing the difference in allelic frequencies between each plate and a pool of the other plates, and by comparing variation in log₁₀ of the autosomal missing call rate in each plate (no significant batch effects were detected); (8) gender identity: based on X chromosome heterozygosity and the means of the intensities of SNP probes on the X and Y chromosomes; (9) autosomal heterozygosity; (10) pair-wise sample relatedness: pair-wise kinship estimates between every subject were computed using PLINK[45]; and (11) genetic ancestry, which was carefully computed by principal component analyses using Eigenstrat[46] and with all European, American,

African and Asian individuals in the 1,000 Genomes Project as the reference (1000GP, phase III).

After QC steps, we removed 66 biologically related participants, 5 participants with unresolved identity issues and 9 outliers of genetic ancestry. We filtered out 27,363 SNPs that failed by CIDR technical filters; 33,152 SNPs with missing call rate $>2\%$; 9,569 SNPs with $>0$ discordant calls in 24 cross-batch duplicates; 684 SNPs with $>1$ discordant calls in 52 within-batch duplicates; 307 SNPs with $>1$ Mendelian error in 8 HapMap trios/duos; 2,939 SNPs with $P < 1 \times 10^{-4}$ in Hardy–Weinberg equilibrium tests; 8,378 SNPs as positional duplicates; and 126,616 monomorphic SNPs. As a result, a total of 2,160,368 SNPs from 1,733 biologically unrelated women were available for subsequent SNP imputation.

**SNP imputation in the discovery stage.** We performed phasing using SHAPEIT[47] and imputation using IMPUTE2 (ref. 48) software, with all individuals in the 1000GP as a reference panel. We applied several post-imputation QC metrics including removal of SNPs with an imputation info score $<0.4$, with a missing call rate $>0.05$, and/or with a MAF $<0.02$. There were 13,317,377 genotyped or imputed SNPs on autosomal chromosomes and/or on the X chromosome available for subsequent data analyses.

**Statistical analyses in the discovery stage.** We first estimated the narrow-sense heritability of overall PTB and spontaneous PTB using the GCTA software[49] based on all genotyped SNPs passing QC and with a MAF $\geq 0.02$. Genome-wide SNP associations with binary (overall PTB and spontaneous PTB) and continuous outcomes (GA) were analysed using logistic and linear regression models, respectively, in PLINK[45] (v1.07) under an additive genetic model (for each genotyped SNP), with adjustment for covariates including maternal age (coded as $<20$, 20–29, 30–34.9 and $\geq 35$ years), genotyping batch, parity (coded as 0, 1, 2 or above), infant's gender and genetic ancestry (represented by the first three principal components from principal component analyses). For each imputed SNP, allelic dosage (of the minor allele) was tested for its associations with PTB outcomes using a frequentist association analysis in SNPTEST (V2.5). The genome-wide significance threshold was set as $P < 5.0 \times 10^{-8}$.

The $G \times E$ analysis was conducted in 1,586 women, after removing 89 women with missing data on pre-pregnancy BMI and 58 women with pre-pregnancy BMI $<18.5$ kg m$^{-2}$ (due to a small sample size in this subgroup). We first tested the genome-wide SNP interactions with the pre-pregnancy BMI category on overall PTB risk using the conventional 1-df interaction test, by adding each SNP (under an additive genetic model), the pre-pregnancy BMI category and their interaction term into the logistic regression model with adjustment of the same covariates described above. The logistic regression model was: logit (Pr (D = 1)) = $b_0 + b_1 \times E(G) + b_2 \times$ BMI_category $+ b_{ge} \times$ BMI_category $\times E(G) + b_c \times C$ (Model A), in which E(G) is the number of copies of the minor allele for genotyped SNPs or the expected dosage of the minor allele for imputed SNPs; BMI_category represents the maternal pre-pregnancy BMI category, which is coded as 0 for 'normal weight' (pre-pregnancy BMI:18.5–24.9 kg m$^{-2}$), 1 for 'overweight' (pre-pregnancy BMI: 25–29.9 kg m$^{-2}$) or 2 for 'obesity' (pre-pregnancy BMI: $\geq 30$ kg m$^{-2}$) and treated as an ordered discrete variable; C represents an array of covariates. BMI_category $\times E(G)$ is the interaction term, and $b_{ge}$ represents the interaction statistical effect. We repeated the analyses by (1) using pre-pregnancy BMI as a quantitative trait; and (2) treating BMI_category as a two-level categorical variable: normal weight versus OWO (which was defined as pre-pregnancy BMI $\geq 25$ kg m$^{-2}$). Similar analyses were performed for different subtypes of PTB. The genome-wide significance threshold was set as $P < 5.0 \times 10^{-8}$.

We then performed the 2-df test proposed by Kraft et al.[33] to search for combined signals of the SNP main effect and interactions with the pre-pregnancy BMI category. We first built a basic model: logit (Pr (D = 1)) = $b_0 + b_2 \times$ BMI_category $+ b_c \times C$ (Model B), and then performed the likelihood ratio test to compare Model A and Model B via 'lrtest' in R, with 2 degrees of freedom. For these genome-wide association analyses, Manhattan plots and quantile–quantile plots were generated using the R package GWASTools[43]. The locuszoom plot was generated using a published web tool[50] and the Hg19/1,000 genome Nov 2014 AFR population as the reference group.

For the identified significant interactions, we used the permutation approach to determine the empirical P value in the discovery sample. Assuming $P_{obs}$ is the P value we observed for the significant interaction finding in our discovery sample, we defined the minimum number of permutations needed as $1/P_{obs}$[51]. A higher number of permutations will provide a more stable estimation of the empirical P value. Therefore, we permutated the PTB case–control status $1.9 \times 10^9$ times for a $P_{obs} = 1.8 \times 10^{-8}$, which is more than $30 \times 1/ P_{obs}$ to ensure a stable empirical P value. We calculated the P value for the interaction term in each permuted sample. The empirical P value for the interaction was then calculated as the proportion of the permutation realizations with a P value for the interaction term $< P_{obs}$.

**Genotyping and data analyses in the replication sample.** Replication analyses were performed in AA women from the two PTB GWAS data sets deposited in dbGaP (the GENEVA study and the GPN study) as described above. Genome-wide genotyping was performed using the HumanOmni1-Quad-v1 chip at CIDR in the

GENEVA study, and using the Affymetrix Genome-wide Human SNP Array 6.0 at the Microarray Facility at the University of Pennsylvania in the GPN study, respectively. After data cleaning using a similar protocol as in the discovery sample, the GENEVA study provided 436 AA mothers (130 PTB cases and 306 TB controls), and the GPN study provided 346 AA mothers (169 TBs and 177 spontaneous PTBs) for replication. The association between each SNP and PTB risk in each replication sample was performed using a similar approach as in the discovery sample, with adjustment of maternal age, genetic ancestry (the first three principal components from the principal component analyses), infant's gender and/or parity (this variable was not available in the GENEVA study).

The GPN study (but not the GENEVA study) had available maternal pre-pregnancy BMI information ($n = 300$ AA mothers with pre-pregnancy BMI $\geq 18.5$ kg m$^{-2}$), which allowed us to replicate our $G \times E$ findings from the BBC. We further investigated whether there were similar $G \times E$ in 683 Caucasian women (with maternal pre-pregnancy BMI $\geq 18.5$ kg m$^{-2}$) from the GPN study. For imputed SNPs in the GPN study (such as rs11161721), the allelic dosage was applied for $G \times E$ analyses. All of these interaction tests were performed using the conventional 1-df test based on the logistic regression model, with the adjustment of the same covariates as described above.

**Meta-analysis on gene-environment interactions.** Genome-wide meta-analysis of gene × pre-pregnancy BMI interaction was performed with the METAL software[52,53] in AA mothers from both the discovery ($N = 1,586$) and the replication cohort ($N = 300$). We first identified 9,929,081 SNPs shared by the two cohorts and extracted the interaction term Wald's test signed Z-statistics for each shared SNP. We carefully aligned the data to ensure that effective and reference allele definitions were consistent across data sets. The overall z-statistic and P value were derived from a weighted sum of the individual statistics. Weights were proportional to the square-root of the number of individuals examined in each cohort and the squared weights summed to 1.

**Gene-environment interaction in mother–infant pairs.** In the BBC, a total of 153 infants (with available maternal pre-pregnancy BMI $\geq 18.5$ kg/m$^2$) were genotyped using the Illumina HumanOmni1-Quad array, the same method as we reported previously in the Chicago Food Allergy Study[54]. Genotyping data at rs11161721 for the 153 infants was extracted and then analysed for its interaction association with maternal pre-pregnancy BMI on overall PTB using the conventional 1-df interaction test based on the logistic regression model.

Among these 153 infants, mothers of 120 infants were included in our discovery sample. To test whether the identified SNP × pre-pregnancy BMI interactions were due to a maternal or paternal effect, we constructed haplotypes for each infant based on the available genome-wide genotyping data in mother–infant pairs, and determined which one came from the mother (and which one from the father). We employed SHAPEIT[47] software to phase the haplotypes around the SNP of interest (rs11161721). The chosen haplotype contained 51 SNPs (25 SNPs upstream of rs11161721 and 25 SNPs downstream), which offered us sufficient resolution in distinguishing the paternal and mother haplotype in the infant. In SHAPEIT, we used the 'duohmm' option to utilize pedigree information in phasing. With the phased haplotype, we could determine which maternal allele at rs11161721 was transmitted. For data analyses, we coded the number of the *rs11161721-A* allele transmitted from the mother (maternal-origin *A*-allele, 1 versus 0) and the number of the *rs11161721-A* allele transmitted from the father (paternal-origin *A*-allele, 1 versus 0) for each infant, and then explored the maternal and paternal effect of rs11161721 × maternal pre-pregnancy BMI interactions by testing the interaction between pre-pregnancy BMI and the maternal-origin *rs11161721-A* allele, as well as between pre-pregnancy BMI and the paternal-origin *rs11161721-A* allele in the same logistic regression model. Similar analyses were conducted in 276 AA mother–infant pairs from the GPN study. Since rs11161721 was not genotyped but imputed in the GPN study, the haplotype was directly provided by the imputation software, IMPUTE2 (ref. 48), and employed to determine allele transmission.

**Functional annotation using existing datasets.** To identify potential causal gene(s) underlying the identified genetic associations with PTB, we queried existing eQTL data sets[30] in omental adipose tissue from 740 subjects and in subcutaneous adipose tissues from 611 subjects to assess whether the identified SNPs were eQTL SNPs. We evaluated the associations between the identified SNP (rs11161721) and transcript expression levels for genes in *cis* and *trans*, where *cis* genes were defined as genes within 500 kb of the SNP or were otherwise defined as *trans* genes. False discovery rates were calibrated for *cis*- and *trans*- associations separately using a permutation procedure.

**Data availability.** The genotyping and phenotypic data in the BBC that support the findings of this study have been archived in dbGaP (entry # phs000332.v3.p2). The two replication data sets from the previous studies were also from dbGaP (the GENEVA study, dbGaP entry #phs000353.v1.p1; and the GPN study, dbGaP entry #phs000714.v1.p1), as we described above in the Methods section.

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

## Acknowledgements

We thank all of the study participants for their support and help with the study. We are also grateful for the dedication and hard work of the field team at the Department of Pediatrics, Boston University School of Medicine, and for the help and support of the obstetric nursing staff at Boston Medical Center. Genotyping services were provided by the CIDR. CIDR is fully funded through a federal contract from the National Institutes of Health to The Johns Hopkins University, contract numbers HSN268200782096C and HHSN268201200008I. The GWAS data cleaning was performed by Dr Laurie and her team at Washington University following the GENEVA protocol. We are grateful for the critical comments from Drs David K Stevenson and Gary M Shaw at the March of Dimes Prematurity Research Center at the Stanford University School of Medicine. The BBC (the parent study) is supported in part by the March of Dimes PERI grants (20-FY02-56, #21-FY07-605), and National Institutes of Health (NIH) grants (R21ES011666, R21HD066471, 2R01HD041702). Dr Tsai is supported by grants from National Science Council (101-2314-B-400-009-MY2) and Ministry of Science and Technology (103-2314-B-400-004-MY3). Dr Ke Hao is partially supported by the National Natural

Science Foundation of China (Grant No's 21477087 and 91643201) and by the Ministry of Science and Technology of China (Grant No. 2016YFC0206507). Dr Hong is partially supported by Hopkins Population Center (NICHD R24HD042854).

## Author contributions

X.W. is the principal investigator of the BBC (the parent study), and has full access to all of the data in the study and takes responsibility for the integrity of the data and the accuracy of the data analysis. The subject recruitment and data collection was overseen by X.W. and conducted by a team of investigators including X.H., C.P., D.C., L.H., S.C., A.L.P, B.Z.; X.H., K.H., X.W. took primary responsibility for study design. C.P., G.W., Y.J., X.L., H.J.T., X.H. prepared samples for genotyping; X.H., S.P., K.H., B.S., A.D.N., H.J.T., X.L., G.M., Z.Z. performed data cleaning and data analyses, with guidance from K.H.,H.J.,D.E.W,M.D.F.,T.H.B., and X.W; X.H, K.H., L.H, T.H.B, M.D.F, A.L.P, D.E.W, S.C., B.Z., and X.W. interpreted the data and the research findings. X.H., K.H., P.S., T.R.B, X.W. wrote and revised the manuscript. All of the co-authors reviewed the manuscript. X.W. obtained the study funding.

## Additional information

**Competing interests:** The authors declare no competing financial interests.

