## [Peer Review File · Nature Communications]

Reviewers' comments:

Reviewer #1 (Remarks to the Author):

Overall this is an interesting paper.

The conclusion that this GxE interaction provides a strategy for "personalized prevention" for PTB is maybe a bit of an overstatement. After all, when a woman gets pregnant, it's too late to modify the pre-pregnancy weight. Also, as pre-pregnancy overweight is a risk factor for PTB no matter the genotype for rs11161721, it is not that we would recommend that only women with a particular genotype would modify their weight - we would recommend that for all!

Keeping this in mind, it would be useful to reorganize the tables somewhat, and provide information of the risk (e.g. OR) of pre-pregnancy OW/OB relative to NW in each of the three genotype strata. That would make it clearer what the personalized medicine implications are.

I am not convinced about the argument that it's comforting that different types of PTB all show the interaction. As each type of PTB uses the same controls and we know already that the subtypes together make all the PTB it would be concerning when this was not the case.

It is not mentioned which LD was used for the locus-zoom plot in Fig 1C. Was it African American (AA) LD (I hope so - that is what should be shown).

Differences in LD between Europeans and AA could be the reason why the AA results did not replicate in the Europeans (though in general we would expect that in the other direction). It would be good to check some of the variants in LD in the Europeans replication sample. A power calculation for the European replication sample would also be useful.

In Table 2 the MAFs are switched for the cases and controls - as they shouldn't differ for the controls between the various sub-groups.

It is not clear whether the babies had just rs11161721 genotyped or that they had a complete GWAS done. If it is the later, you should be able to construct haplotypes for the babies, and determine which one comes from the mother (and which one from the father). If that is the case, you can directly test the maternal and paternal effect of the SNP (or GxE), and you wouldn't need to make the argument that it's likely a maternal effect because the effect is smaller when you test the baby's genotype.

Reviewer #2 (Remarks to the Author):

Hong and colleagues have performed a genome-wide association study of preterm birth in mothers of 698 preterm infants and mothers of 1035 infants born at term. They searched for associations between individual SNPs and PTB. They additionally tested for GxE interaction, assessing whether SNP-PTB associations were different in pre-pregnancy BMI categories of "normal" vs overweight vs obese women. Of three SNP associations identified in the overall analysis at $P < 5 \times 10^{-8}$, two did not replicate in the independent GENEVA or GPN studies, while the other SNP was not available in replication studies. The SNP-pre-pregnancy BMI interaction analysis showed evidence that association between rs11161721 (in COL24A1) and PTB differed between BMI groups ($P = 1.7 \times 10^{-8}$). This association was consistently observed when using different PTB definitions or when examining

gestational age as a quantitative trait. The SNP was also associated with expression of COL24A1 in adipose tissue. Using available data from the (smaller) GPN African-American study, a consistent association was observed ($P=0.01$). This was not observed in the Caucasian replication study.

This is an important area of research and the findings are novel. The manuscript is clear and well written. This study has been performed thoroughly, with a sensible choice of analyses and with attempts to replicate the key findings from the discovery stage. The authors have done a very good job using the data available to them. The result is intriguing, and the replication and eQTL data help to strengthen the initial evidence from the GWAS. However, there are some important areas of concern.

1. The major concern is that overall power is low, especially for a GxE analysis. Can the authors justify the choice of the $P<5e-08$ threshold and explain why a lower threshold was not required for the GxE analysis?

2. While replication in an additional African American cohort strengthens the evidence somewhat, the replication study is very small. There was no replication in the Caucasian cohort. While an ethnicity-specific effect/residual confounding (as noted by the authors) are possible explanations for this discrepancy, a false positive result is also a possibility. The lack of replication in the Caucasian study should be mentioned in the abstract and the possibility of a false positive result should be acknowledged in the discussion.

3. Introduction: "While several genome-wide association studies... have been conducted...none of them have identified significant maternal genetic variants for PTB." What is the maximum size of GWAS published to date? Could it be that the "missing heritability" for PTB is largely due to variants with effect sizes that are too small to have been detectable in studies to date? I think this is quite a likely explanation, but it is not fully appreciated in the text.

4. In the results section (text), the results for imputed SNPs are presented separately after the results of genotyped SNPs. It took me a little while to understand why there was a separation and it made reading the results a little confusing. If the SNPs are well imputed, there should not be any major reason to present the results separately. Since the imputed SNPs are annotated in the tables, it would be clearer in the text to present results for all SNPs that showed associations in the discovery data together, followed by the results for SNPs available in replication studies (with an explanation for why some were not available).

5. Would it be possible to run a GWAS meta-analysis of all SNPs available in the current study and in the dbGAP studies? Perhaps even a trans-ancestry GWAS meta-analysis? This would have greatly improved power over a much larger number of SNPs than the present approach of looking up top hits.

6. In table 1 it would be helpful to the reader to present the mean (SD) age in cases and in controls instead of counts available in the various groups (like has been done for gestational age). For BMI, the mean (SD) in cases and in controls could be presented in addition to the counts in the groups.

Reviewer #3 (Remarks to the Author):

This is a well written manuscript that reports a GWAS study of preterm birth (PTB) in African American women. The authors identify one intronic SNP (rs11161721) in the COL24A1 gene that associates with PTB and was replicated in a validation cohort. They go on to examine gene environment interactions (G x E) between the SNP and maternal BMI and risk for PTB. Their original hypothesis was that G x E are responsible for the inconsistency in reports of maternal BMI associations with PTB. However, in their own cohort the authors report that overweight and obesity was higher in women who delivered preterm than term but this was not actually statistically significant so they should report the

percentages and indicate these are similar. The wording around this needs some rewriting. Also the whole manuscript is framed with the idea that obesity is an important risk factor for PTB. I think this needs to be toned down. It is clear that there are different associations of the SNP with PTB when considering lean versus obese women. The G x E data are very interesting and will be of interest to many others. As far as I know this is the first time that anyone has looked at this specifically for maternal BMI. The abstract accurately summarises the research. The statistical methods are appropriate and the supplementary data are well presented on the whole.

It is not always clear when the cases are all PTB or spontaneous PTB. As the authors point out all PTB includes iatrogenic PTB for a number of medical indications including preeclampsia. This would be a massive confounder in the analyses and would likely be a big contributor to the variation in the literature as to the association of BMI with PTB. Please ensure that all PTB and SPTB are explicitly stated throughout the manuscript and the supplementary tables and figures. Note spontaneous is incorrectly spelt on the heading of Suppl figure 1B. This may change the reader's understanding of the data.

Although the authors have shown that COL24A1 rs11161721 affects expression in adipose tissue, I am unsure how collagen deposition in adipose tissue could be related to that in the fetal membranes. This is particularly because the fetal membranes express just one maternal allele for each gene and that of the father is the other. Quite rightly the authors point out in their introduction that it is only maternal genotypes and not paternal genotypes that have been associated with PTB. Some discussion of this could be useful. Ideally it would be good to also use fetal genotypes in the study but I know that these are often not available. See York et al. 2013 AmJOG.

"intra-utero inflammation" should be "intra-uterine inflammation"

In the discussion the authors state "In other words, the impact of pre-pregnancy BMI on PTB risk depended on the maternal genotype at rs11161721." An alternative might be that the impact of the SNP is dependent on maternal BMI since in lean women the SNP was associated with an increase in PTB but in obese women it was lower than wildtype. This is consistent with the notion that for polygenic syndromes the impact of individual SNPs is quite modest. There are plenty of Caucasian pregnant women who are overweight and obese. A couple of lines on why the associations were not found in Caucasian women would be helpful.

Once one environmental interaction has been included it begs the question why others that are known to associate with PTB are not also included.

Reviewers' comments:

Reviewer #1 (Remarks to the Author):

I think that my comments are well addressed.

Reviewer #2 (Remarks to the Author):

The authors have done a thorough job, responding to the reviewers' comments. My major reservation remains, however, about the sample size and power for replication, especially considering that this is searching for G*E and not individual SNP effects. On balance, the study is not sufficiently powered to yield a strong conclusion. The reader would be much more confident in the result if the authors were to be able to include additional replication data.

Reviewer #4 (Remarks to the Author):

This is an interesting paper investigating a relatively less studied genetic component of preterm birth. As the authors pointed out, studies collecting such data are few and relatively small. The manuscript is much improved with the reviewers comments, and worth pointing out that the replication of GxE finding in the same ethnic group does strengthen the discovery.

- Genome-wide GxE has been difficult due to the heterogeneity in environmental exposures, and lack of replication across studies. In this study, the issue is compounded with small sample sizes. Several methods have been used for assessing the significance of GxE interactions: conventional Bonferroni threshold, permutation, family-wise discovery rate (though the Bonferroni p-value threshold is still by far most commonly used). Suggest the authors permute the case control labels for PTB to get an empirical p-value for the discovery study on top of the meta-analysis with the replication study. See recent paper in PLoS Genetics 27723779.

- I also have some difficulty with the two revised tables: Table 2 and 3. They are a little complicated, and does not allow for quick comparison across the different strata compared to the reference group. Figure 2 is nicely done, and allows for immediate visual assessment of the interaction and differences in the BMI categories across genotypes. Since genotypes are fitted on additive scale, the tables can be simplified to present the main effects of each A allele (for Table 3), overweight/obese and interaction. Suggest to see recent paper in PLoS Genetics 27723779 for simplified table representation.

- I agree with reviewer one that this finding is still immature for personalised prevention, especially with the power calculations presented for the Caucasians.

- Results section can be more concise. The section on sensitivity of the variant rs11161721, the first paragraph may be shortened to only refer to Figure 2. The OR comparison in stratified genotype classes in Table 2 could be moved to Supplementary. It may also be more accurate to say "consistency of interaction across PTB subtypes"

REVIEWERS' COMMENTS:

Reviewer #4 (Remarks to the Author):

It is reassuring that the same conclusions are reached with permutations. Tables have been simplified and made more readable. Note font size of Table 2 footnotes. As with many genome-wide GxE and replications, authors have made clear the limitations, including less strong conclusions on the utility of current findings in personalised medicine. The authors have addressed all my concerns adequately.

Point-to-point responses

Manuscript No # NCOMMS-16-07795

Title: Genome-wide approach identified a novel gene-maternal pre-pregnancy BMI interaction on preterm birth

Point-to-point responses to Reviewer #1

1. **“Overall this is an interesting paper.**

The conclusion that this GxE interaction provides a strategy for "personalized prevention" for PTB is maybe a bit of an overstatement. After all, when a woman gets pregnant, it's too late to modify the pre-pregnancy weight. Also, as pre-pregnancy overweight is a risk factor for PTB no matter the genotype for rs11161721, it is not that we would recommend that only women with a particular genotype would modify their weight - we would recommend that for all! ”

Response: We have revised our conclusion as follows: “If further validated, these findings may provide new insight into the etiology of PTB and improve our ability to predict and prevent PTB” [Page 15, Paragraph 2 of the marked manuscript]. For example, knowing a woman’s BMI and genotype may help her health care providers to take active measures to optimize her weight before pregnancy and her weight gain during pregnancy.

2. **“Keeping this in mind, it would be useful to reorganize the tables somewhat, and provide information of the risk (e.g. OR) of pre-pregnancy OW/OB relative to NW in each of the three genotype strata. That would make it clearer what the personalized medicine**

implications are.”

Response: We have revised the tables (Table 2, Table 3, and Supplementary Tables 4 - 6) accordingly to provide information regarding the risk of pre-pregnancy OW/OB relative to normal weight (NW) in each of the three genotype strata at rs11161721. To be consistent, Figure 2 was also reorganized. The relevant text in the Result section has also been revised accordingly, as shown below:

“**Table 2** presents the ORs of pre-pregnancy BMI category (coded as 0=normal weight, 1=overweight, 2=obesity) in each of the three *rs11161721* genotype strata. A one-strata increase in pre-pregnancy BMI category was associated with a 1.4 fold (95%CI=1.2-1.7) higher risk of overall PTB in mothers carrying the *rs11161721-CC* genotype; however, it was associated with a lower risk in mothers carrying the *rs11161721-CA* genotype (OR=0.8, 95%=0.6-1.0) as well as in mothers carrying the *rs11161721-AA* genotype (OR=0.4, 95%CI=0.2-0.8). The joint associations between *rs11161721* and pre-pregnancy BMI are presented in **Figure 2.**” [Page 7, Paragraph 4 of the marked manuscript]

3. “I am not convinced about the argument that it's comforting that different types of PTB all show the interaction. As each type of PTB uses the same controls and we know already that the subtypes together make all the PTB it would be concerning when this was not the case.”

Response: We have revised the related sentence as follows: “We examined if the effect of *rs11161721*×pre-pregnancy BMI interaction on overall PTB varied by subtypes of PTB, that is, if there was heterogeneity in the magnitude and direction of the interaction effect across groups of preterm infants (**Table 2**). We found the interaction effect size and direction between *rs11161721* and pre-pregnancy BMI was comparable across all subtypes of PTB (**Table 2**), including

spontaneous PTB ($P_{G \times E} = 1.2 \times 10^{-5}$), medically-indicated PTB ($P_{G \times E} = 1.7 \times 10^{-5}$), early PTB ($P_{G \times E} = 0.003$), late PTB ($P_{G \times E} = 1.3 \times 10^{-7}$), PTB with ($P_{G \times E} = 3.0 \times 10^{-4}$) and without intra-uterine inflammation (IUI, $P_{G \times E} = 7.4 \times 10^{-7}$), and GA as a continuous outcome ($P_{G \times E} = 3.5 \times 10^{-5}$).

” [Page 8, Paragraph 2 of the marked manuscript].

We also updated our Discussion accordingly:

“Our study suggests the magnitude and direction of the *rs11161721* × pre-pregnancy BMI interaction associations were comparable across PTB subtypes, including both spontaneous and medically-indicated PTB, despite the fact that these two subtypes may have distinct clinical features.” [Page 11, Paragraph 3 of the marked manuscript]

4. “ It is not mentioned which LD was used for the locus-zoom plot in Fig 1C. Was it African American (AA) LD (I hope so - that is what should be shown).”

Response: In Fig 1C, the LD was calculated using Hg19/1000 genome Nov 2014 AFR population. We have added this information into the revised Methods as follows:

“The locuszoom plot was generated using a published web tool⁵⁰ and the Hg19/1000 genome Nov 2014 AFR population as the reference group.” [Page 21, Paragraph 1 of the marked manuscript]

5. “ Differences in LD between Europeans and AA could be the reason why the AA results did not replicate in the Europeans (though in general we would expect that in the other direction). It would be good to check some of the variants in LD in the Europeans replication sample. A power calculation for the European replication sample would also be useful.”

Response: In the revision, we calculated LD between *rs11161721* and nearby SNPs (within

1Mb) in African American mothers from the BBC. There are 7 SNPs in moderate to high LD ($r^2 > 0.4$) with rs11161721. Each of these 7 SNPs showed either significant or suggestive interaction with pre-pregnancy BMI category in associations with PTB in the discovery sample. However, none of these interaction effects was identified in Caucasian mothers from the GPN cohort (all p for interaction > 0.05).

As suggested, we investigated the LD structure for the region within 1MB of rs11161721 in African Americans and in Caucasians, using the genotyped data from the discovery (BBC) and the replication (GPN) cohorts, respectively. We found the LD structure in African Americans from the GPN study (Fig 1A, see below) was comparable with that in our discovery cohort (Fig 1B, see below). The top SNP, rs11161721, is located in the LD block marked with the blue bar. This LD block in Caucasians from the GPN study (Fig 1C) is somewhat wider than that for African Americans, which is expected. Among the 683 Caucasian women from the GPN study, we then tested for interactions between pre-pregnancy BMI \times SNP genotype on PTB for all of the SNPs in the same LD block with rs11161721. No significant interaction was found (Fig 1d). Taking these findings together, our data indicate that the lack of replication in the Caucasian cohort for the African American results is unlikely due to differences in LD between Europeans and African Americans.

Figure 1. The LD structure of the *COL24A1* gene in African Americans and in Caucasians, and the locuszoom plot for the interaction between pre-pregnancy BMI category and SNPs in the same LD block with rs11161721 in Caucasians. Fig 1a-c are the LD structures in African-American women from the GPN study (1A), from the BBC (1B) and in Caucasians from the GPN study (1C), respectively. 1D) is the locuszoom plot for the interaction between pre-pregnancy BMI category and SNPs in the same LD block with rs11161721 in Caucasian women from the GPN study. In this plot, the LD between each SNP and rs11161271 is calculated using hg19/1000 Genome Nov 2014 EUR population as the reference group.

We then performed a power calculation for the Caucasian replication sample using “Quanto” software. Based on the current data, we assumed that 1) the minor allele frequency of rs11161721 is 0.28; 2) the proportion of overweight/obesity is 35.5%; 3) the marginal effect of rs11161721 and pre-pregnancy overweight/obesity is 1.0 and 1.38 (estimated based on the current available data), respectively; and 4) we used the conventional threshold for significance ($p < 0.05$); this Caucasian replication sample (including 365 TB mothers and 318 PTB mothers) had $>80\%$ power to identify an interaction $OR \leq 0.48$ (the interaction OR was 0.5 and 0.37, respectively, in our BBC discovery samples and the GPN African-American samples), but the power would be $<50\%$ if the interaction OR was > 0.61 . Based on these results, we suggest that the failure to replicate the interaction effect in Caucasians may be due to multiple reasons, as we have summarized in the revised Discussion:

“Finally, our finding on the interaction effect between pre-pregnancy BMI and *rs11161721* (as well as other SNPs in LD with *rs11161721*) was not observed in Caucasian mothers of the GPN study, which may have been due to the following reasons: 1) there were some residual confounding factors not considered in our analyses of Caucasian mothers; 2) the interaction effect size in Caucasians was relatively modest, and thus the current study in 683 Caucasian mothers

would have limited power to identify such an interaction effect; 3) the identified interaction effect may be specific to African Americans, and/or 4) we still could not rule out the possibility that our finding in African Americans is a false positive result, although such chance is low because the finding has been validated in a totally independent cohort. Further replications in larger African-American and Caucasian cohorts are still needed.” [Page 14, Paragraph 2 of the marked manuscript].

6. “In Table 2 the MAFs are switched for the cases and controls - as they shouldn't differ for the controls between the various sub-groups.”

Response: In the revision, we have updated Table 2 according to comment #2 above, and the OR (95%CI) of pre-pregnancy OW/OB relative to NW is now shown for each of the three genotype strata.

7. “It is not clear whether the babies had just rs11161721 genotyped or that they had a complete GWAS done. If it is the later, you should be able to construct haplotypes for the babies, and determine which one comes from the mother (and which one from the father). If that is the case, you can directly test the maternal and paternal effect of the SNP (or GxE), and you wouldn't need to make the argument that it's likely a maternal effect because the effect is smaller when you test the baby's genotype.”

Response: As suggested, we constructed haplotypes for the 120 African-American infants from the discovery cohort and for the 276 African-American infants from the replication cohort, based on the available genome-wide genotyping data for the mother-infant pairs, and determined which one came from the mother (and which one from the father). We employed SHAPEIT software to phase haplotype around the SNP of interest (rs11161721). The haplotype was chosen to contain 51 SNPs (25 SNPs upstream of the SNP of interest and 25 SNPs downstream), which offered us

sufficient resolution to distinguish the paternal and mother haplotypes in the infant. In SHAPEIT, we used the “duohmm” option to utilize pedigree information in phasing. With the phased haplotype, we could determine which maternal allele at rs11161721 was transmitted. For data analyses, we coded the number of the rs11161721-A allele transmitted from the mother (maternal-origin A allele, 1 vs 0) and the number of the rs11161721-A allele transmitted from the father (paternal-origin A allele, 1 vs 0) for each infant, and then explored the maternal and paternal effect of rs11161721 × maternal pre-pregnancy BMI interactions by testing the interaction between pre-pregnancy BMI and the maternal-origin rs11161721-A allele, as well as between pre-pregnancy BMI and the paternal-origin rs11161721-A allele in the same logistic regression model. Similar analyses were conducted in 276 African American mother-infant pairs from the GPN study. Since rs11161721 was not genotyped but imputed in the GPN study, the haplotype was directly provided by the imputation software, IMPUTE2, and employed to determine allele transmission. This information has been added to the revised Methods [**Page 22, Paragraph 3 of the marked manuscript**]

The relevant results are presented in **Supplementary Table 7** and in the revised Results section as follows:

“With available GWAS data in 120 mother-infant pairs from the BBC and in 276 mother-infant pairs from the GPN study, we coded the number of maternal-origin *rs11161721-A* allele and paternal-origin *rs11161721-A* allele for each infant (see Methods), and tested their interactions with pre-pregnancy BMI category on overall PTB. Among the 120 AA infants from the BBC, the observed effect size for the maternal pre-pregnancy BMI × maternal-origin *rs11161721-A* allele interaction was higher than that for the maternal pre-pregnancy BMI × paternal-origin *rs11161721-A* allele interaction, although neither was statistically significant. In comparison, among the 276 AA infants from the GPN study, the interaction between pre-pregnancy BMI category and maternal-origin *rs11161721-A* allele was statistically significant ($P_{G \times E} = 0.004$), while no significant interaction was found between pre-pregnancy BMI and the paternal-origin

rs11161721-A allele, suggesting a significant maternal effect (Supplementary Table 7).[Page 10, Paragraph 2 of the marked manuscript].

Point-to-point responses to comments from Reviewer #2

1. “Hong and colleagues have performed a genome-wide association study of preterm birth in mothers of 698 preterm infants and mothers of 1035 infants born at term. They searched for associations between individual SNPs and PTB. They additionally tested for GxE interaction, assessing whether SNP-PTB associations were different in pre-pregnancy BMI categories of "normal" vs overweight vs obese women. Of three SNP associations identified in the overall analysis at $P < 5 \times 10^{-8}$, two did not replicate in the independent GENEVA or GPN studies, while the other SNP was not available in replication studies. The SNP-pre-pregnancy BMI interaction analysis showed evidence that association between *rs11161721* (in *COL24A1*) and PTB differed between BMI groups ($P = 1.7 \times 10^{-8}$). This association was consistently observed when using different PTB definitions or when examining gestational age as a quantitative trait. The SNP was also associated with expression of *COL24A1* in adipose tissue. Using available data from the (smaller) GPN African-American study, a consistent association was observed ($P = 0.01$). This was not observed in the Caucasian replication study.

This is an important area of research and the findings are novel. The manuscript is clear and well written. This study has been performed thoroughly, with a sensible choice of analyses and with attempts to replicate the key findings from the discovery stage. The authors have done a very good job using the data available to them. The result is intriguing, and the replication and eQTL data help to strengthen the initial evidence from the GWAS.”

Response: We thank the reviewer for this positive and very insightful comment. As detailed below, we have performed additional analyses to further strengthen our findings.

2. “However, there are some important areas of concern.

1. The major concern is that overall power is low, especially for a GxE analysis. Can the authors justify the choice of the $P < 5e-08$ threshold and explain why a lower threshold was not required for the GxE analysis?”

Response: In the revision, we further acknowledged in the Discussion that “the statistical power of this study may be limited, especially for those SNPs with a low minor allele frequency (i.e., $MAF < 10\%$) and/or SNPs with relatively modest interaction effects with pre-pregnancy BMI” [Page 14, Paragraph 2 of the marked manuscript], and that “we still could not rule out the possibility that our finding in African Americans is a false positive result.” [Page 14, Paragraph 2 of the marked manuscript].

Since there is no well-established p-value threshold for genome-wide G×E interaction study, we adopted the significance threshold ($P < 5 \times 10^{-8}$) that is well-accepted for genome-wide association studies, which is calculated based on the estimated number of independent tests in the genome if all common SNPs in HapMap were tested with direct genotyping or imputation (i.e., $\sim 10^6$ tests, at $\alpha = 0.05$ level). A similar number of independent tests were performed in our G×E analysis.

To minimize false positive and increase statistical power, we applied a two-stage (discovery stage and replication stage) approach in this study, and only reported those signals that can be replicated in an independent population. We also performed a meta-analysis on SNP-BMI interaction to combine the associations identified in the discovery and the replication samples, and found that the rs11161721×(pre-pregnancy BMI) interaction became more significant, with $P_{G \times E} = 3.6 \times 10^{-9}$. These results are reported on Page 9, Paragraph 2 of the marked manuscript.

3. “While replication in an additional African American cohort strengthens the evidence somewhat, the replication study is very small. There was no replication in the Caucasian cohort. While an ethnicity-specific effect/residual confounding (as noted by the authors) are possible explanations for this discrepancy, a false positive result is also a possibility. The lack of replication in the Caucasian study should be mentioned in the abstract and the possibility of a false positive result should be acknowledged in the discussion. ”

Response: In the abstract, we have added a statement that these findings were not replicated in Caucasians. In the revised Discussion, we acknowledged that although our findings were replicated in an independent AA cohort, we failed to replicate the findings in Caucasian mothers from the GPN study, which may have been due to the multiple reasons (see our response to comment #5 from Reviewer #1). Specifically, we have now noted that “...we still could not rule out the possibility that our finding in African Americans is a false positive result.” [Page 14, Paragraph 2 of the marked manuscript]

4. Introduction: "While several genome-wide association studies... have been conducted...none of them have identified significant maternal genetic variants for PTB." What is the maximum size of GWAS published to date? Could it be that the "missing heritability" for PTB is largely due to variants with effect sizes that are too small to have been detectable in studies to date? I think this is quite a likely explanation, but it is not fully appreciated in the text.

Response: We have revised the Introduction accordingly, as follows:

“At least three possible explanations for “missing heritability” have been proposed and tested: (1) genetic variants with effect sizes that are too small to have been detectable in studies to date (i.e.,

<5000 mothers or infants in the current GWAS of PTB).” [Page 3, Paragraph 3 of the marked manuscript]

5. “In the results section (text), the results for imputed SNPs are presented separately after the results of genotyped SNPs. It took me a little while to understand why there was a separation and it made reading the results a little confusing. If the SNPs are well imputed, there should not be any major reason to present the results separately. Since the imputed SNPs are annotated in the tables, it would be clearer in the text to present results for all SNPs that showed associations in the discovery data together, followed by the results for SNPs available in replication studies (with an explanation for why some were not available).”

Response: As recommended, in the revised manuscript we have presented the results for all SNPs together (both genotyped and imputed) that showed associations with PTB outcomes in the discovery sample,

“In the discovery sample, we identified *rs149014416*, an imputed SNP near 8p12, was genome-wide significantly associated with overall PTB ($P=1.1\times 10^{-8}$, **Supplementary Table 3a & Supplementary Fig. 1a**). Mothers carrying one copy of *rs149014416-A* allele were at a 2.3 (95%CI, 1.7-3.2) times higher risk of having a PTB baby (**Supplementary Table 3a**). A similar association was found for this SNP when spontaneous PTB was analyzed as the outcome ($P=1.7\times 10^{-8}$). We also found that *rs1558001* at 7q21-22 for spontaneous PTB and *rs8029754* at 15q26 for gestational age at delivery (GA) were both genome-wide significant (**Supplementary Fig. 1**). ” [Page 6, Paragraph 2 of the marked manuscript]

“The two replication cohorts, the GENEVA and the GPN studies (see Methods), both had genotype data for *rs1558001* and *rs8029754*. However, these two SNPs showed no associations with PTB outcomes in AA mothers from these replication cohorts, and the estimated ORs were

not comparable to those in our discovery sample (**Supplementary Table 3b**). SNP *rs149014416*, which failed to be imputed in the two replication cohorts, was dropped from these analyses.

.” [Page 6, Paragraph 3 of the marked manuscript]

We also re-ran the G×E interactions for all of the genotyped and/or imputed SNPs, and updated Figure 1 accordingly. Briefly, the results remain unchanged.

6. “Would it be possible to run a GWAS meta-analysis of all SNPs available in the current study and in the dbGAP studies? Perhaps even a trans-ancestry GWAS meta-analysis? This would have greatly improved power over a much larger number of SNPs than the present approach of looking up top hits.”

Response: In our application for the pertinent datasets from dbGaP, we aimed to 1) replicate our GWAS findings in African-American mothers; 2) examine whether fetal genes may act independently or through interaction with maternal genes to affect the risk of PTB; and 3) explore whether the findings in African-Americans can be extended to other race/ethnicities. As part of our agreement with dbGaP, we promised not to perform meta-analyses for standard GWAS of PTB using the dbGaP data, therefore we respected these limitations fully throughout the course of our study.

In the revision, we ran a genome-wide meta-analysis on SNP × pre-pregnancy BMI interaction for all the genotyped and/or imputed variants available in both the discovery and the replication AA women (N=1,886). We found that rs11161721 remained the top SNP yielding a genome-wide significant interaction with pre-pregnancy BMI category ($P_{G \times E} = 3.6 \times 10^{-9}$). SNPs rs1324899, rs60891279 and rs10443169 in the *COL24A1* gene, which were all in moderate LD with rs11161721 ($R^2 > 0.4$), also met the significance threshold. The Manhattan plot for the meta-analyses is shown in Supplementary Fig 3. The results are reported on **Page 9, Paragraph 2 of the marked manuscript**.

The Methods section has also been revised accordingly and as follows:

“Meta analyses on gene-environment interactions

Genome-wide meta-analysis of gene \times pre-pregnancy BMI interaction was performed with the METAL software^{51,52} in AA mothers from both the discovery (N=1,586) and the replication cohort (N=300). We first identified 9,929,081 SNPs shared by the two cohorts and extracted the interaction term Wald’s test signed Z-statistics for each shared SNP. We carefully aligned the data to ensure that effective and reference allele definitions were consistent across datasets. The overall z-statistic and p-value were derived from a weighted sum of the individual statistics. Weights were proportional to the square-root of the number of individuals examined in each cohort and the squared weights summed to 1.” [Page 22, Paragraph 1 of the marked manuscript]

7. “In table 1 it would be helpful to the reader to present the mean (SD) age in cases and in controls instead of counts available in the various groups (like has been done for gestational age). For BMI, the mean (SD) in cases and in controls could be presented in addition to the counts in the groups.”

Response: We have updated Table 1 as suggested.

Point-to-point responses to comments from Reviewer #3

1. “This is a well written manuscript that reports a GWAS study of preterm birth (PTB) in African American women. The authors identify one intronic SNP (rs11161721) in the COL24A1 gene that associates with PTB and was replicated in a validation cohort. They go on to examine gene environment interactions (G \times E) between the SNP and maternal BMI and risk for PTB. Their original hypothesis was that G \times E are responsible for the

inconsistency in reports of maternal BMI associations with PTB. However, in their own cohort the authors report that overweight and obesity was higher in women who delivered preterm than term but this was not actually statistically significant so they should report the percentages and indicate these are similar. The wording around this needs some rewriting. Also the whole manuscript is framed with the idea that obesity is an important risk factor for PTB. I think this needs to be toned down. It is clear that there are different associations of the SNP with PTB when considering lean versus obese women. ”

Response: As suggested, we have revised the Introduction, Results and Discussion sections as follows:

“2) Previous studies on the associations of pre-pregnancy BMI or OWO with PTB have yielded inconsistent results, including positive²⁰⁻²³, null^{24,25} or negative associations.²⁶⁻²⁸. It is possible that such inconsistent findings may be in part due to gene × maternal pre-pregnancy BMI interaction²⁹, which is largely unexplored; ” **[Page 4, Paragraph 2 of the marked manuscript]**

“The rates of pre-pregnancy OWO were 53.4% in PTB cases and 48.1% in TB controls ($P=0.10$).” **[Page 5, Paragraph 2 of the marked manuscript]**

“ Our findings help to explain the inconsistent findings between pre-pregnancy BMI and PTB outcomes reported in previous studies²⁰⁻²⁸ and suggest such inconsistency may be partly due to the interactions between maternal genotype (i.e., rs11161721) and pre-pregnancy BMI. We showed the impact of pre-pregnancy BMI on overall PTB risk depends on the maternal genotype at rs11161721. Notably, the risk of overall PTB in obese mothers carrying the CC genotype was about two times higher than in their normal-weight counterparts, but the risk decreased by 50% in obese mothers carrying the AA genotype. An alternative explanation might be that the impact of the SNP depends on maternal pre-pregnancy BMI, since in normal-weight women the A-allele at

rs11161721 was associated with a higher risk of having overall PTB but in obese women it was associated with a lower risk compared to the C-allele.” [Page 11, Paragraph 2 of the marked manuscript]

2. “The G x E data are very interesting and will be of interest to many others. As far as I know this is the first time that anyone has looked at this specifically for maternal BMI. The abstract accurately summarises the research. The statistical methods are appropriate and the supplementary data are well presented on the whole.”

Response: We appreciate the positive comments about our manuscript.

3. “It is not always clear when the cases are all PTB or spontaneous PTB. As the authors point out all PTB includes iatrogenic PTB for a number of medical indications including preeclampsia. This would be a massive confounder in the analyses and would likely be a big contributor to the variation in the literature as to the association of BMI with PTB. Please ensure that all PTB and SPTB are explicitly stated throughout the manuscript and the supplementary tables and figures. Note spontaneous is incorrectly spelt on the heading of Suppl figure 1B. This may change the reader's understanding of the data.”

Response: In the revised manuscript, we have ensured that overall PTB and spontaneous PTB are presented clearly. We have also corrected the spelling error. We highlighted that, despite the fact that spontaneous and medically indicated PTB are heterogeneous entities, the interaction between *rs11161721* and pre-pregnancy BMI on PTB appears comparable for overall PTB, spontaneous PTB, and medically indicated PTB. (See revised **Table 2**)

4. “Although the authors have shown that COL24A1 rs11161721 affects expression in adipose tissue, I am unsure how collagen deposition in adipose tissue could be related to that in the fetal membranes. This is particularly because the fetal membranes express just one maternal allele for each gene and that of the father is the other. Quite rightly the authors point out in their introduction that it is only maternal genotypes and not paternal genotypes that have been associated with PTB. Some discussion of this could be useful. Ideally it would be good to also use fetal genotypes in the study but I know that these are often not available.

Response: In the revision, we have tested the maternal and paternal effect of the identified G×E (see our response to Comment # 7 from Reviewer #1). Briefly, with available GWAS data in 120 mother-infant pairs from the BBC and in 276 mother-infant pairs from the GPN study, we coded the number of maternal-origin *rs11161721-A* allele and paternal-origin *rs11161721-A* allele for each infant (see Methods), and tested their interactions with pre-pregnancy BMI category on overall PTB. Among the 120 AA infants from the BBC, the observed effect size for the maternal pre-pregnancy BMI × maternal-origin *rs11161721-A* allele interaction was higher than that for the maternal pre-pregnancy BMI × paternal-origin *rs11161721-A* allele interaction, although neither were statistically significant. In comparison, among the 276 AA infants from the GPN study, the interaction between pre-pregnancy BMI category and maternal-origin *rs11161721-A* allele was statistically significant ($P_{G \times E} = 0.004$) while no significant interaction was found between pre-pregnancy BMI and the paternal-origin *rs11161721-A* allele, suggesting a significant maternal effect (**Supplementary Table 7**). [Page 10, Paragraph 2 of the marked manuscript]

We have also added the following paragraph to the revised Discussion:

“The fetal membranes are known to express one maternal allele and one paternal allele for each genetic variant. Our analyses using the AA mother-infant pairs from the replication

cohort indicated that the *rs11161721*×pre-pregnancy BMI interaction is due to a maternal rather than a paternal effect. Consistently, previous studies have also shown that maternal genetic factors, but not paternal genetic factors, contribute significantly to PTB risk⁸. These findings support the use of genetic studies that focus more on the maternal genome and place less emphasis on collecting data on paternal genes.” [Page 12, Paragraph 2 of the marked manuscript]

5. “See York et al. 2013 AmJOG.

"intra-utero inflammation" should be "intra-uterine inflammation"

Response: We have revised this phrasing accordingly.

6. “ In the discussion the authors state "In other words, the impact of pre-pregnancy BMI on PTB risk depended on the maternal genotype at *rs11161721*." An alternative might be that the impact of the SNP is dependent on maternal BMI since in lean women the SNP was associated with an increase in PTB but in obese women it was lower than wildtype. This is consistent with the notion that for polygenic syndromes the impact of individual SNPs is quite modest.

Response: We have revised the Discussion accordingly, as shown below:

“We showed that the impact of pre-pregnancy BMI on overall PTB risk depends on the maternal genotype at *rs11161721*. Notably, the risk of overall PTB in obese mothers carrying the CC genotype was about two times higher than in their normal-weight counterparts, but the risk decreased by 50% in obese mothers carrying the AA genotype. An alternative explanation might be that the impact of the SNP depends on maternal pre-pregnancy BMI, since in normal-weight women the A-allele at *rs11161721* was associated with a higher risk of having overall PTB but in

obese women it was associated with a lower risk compared to the C-allele.” [Page 11, Paragraph 2 of the marked manuscript]

7, “There are plenty of Caucasian pregnant women who are overweight and obese. A couple of lines on why the associations were not found in Caucasian women would be helpful.”

Response: We have added several sentences to the revised Discussion to explain why the interaction was not found in Caucasians, as shown below,
“Finally, our finding on the interaction effect between pre-pregnancy BMI and *rs11161721* (as well as other SNPs in LD with *rs11161721*) was not observed in Caucasian mothers of the GPN study, which may have been due to the following reasons: 1) there were some residual confounding factors not considered in our analyses of Caucasian mothers; 2) the interaction effect size in Caucasians was relatively modest, and thus the current study in 683 Caucasian mothers would have limited power to identify such an interaction effect; 3) the identified interaction effect may be specific to African Americans, and/or 4) we still could not rule out the possibility that our finding in African Americans is a false positive result, although such chance is low because the finding has been validated in a totally independent cohort. Further replications in larger African-American and Caucasian cohorts are still needed.” [Page 14, Paragraph 2 of the marked manuscript]

8. “Once one environmental interaction has been included it begs the question why others that are known to associate with PTB are not also included.”

Response: In this study, we focused on pre-pregnancy BMI for the following reasons (Page 4, Paragraph 2 of the marked manuscript): “1) Maternal pre-pregnancy overweight/obesity (OWO) is quite prevalent in the BBC (>50%), which is significant from clinical and public

health perspectives. The high prevalence of pre-pregnancy OWO also ensures sufficient statistical power to identify a significant G×E in this study; 2) Previous studies on the associations of pre-pregnancy BMI or OWO with PTB have yielded inconsistent results, including positive²⁰⁻²³, null^{24,25} or negative associations.²⁶⁻²⁹ It is possible that such inconsistent findings may be in part due to maternal gene × pre-pregnancy BMI interaction,³⁰ which is largely unexplored; and 3) Maternal pre-pregnancy OWO is potentially modifiable, and such relevant studies may help women and health care providers to take active measures to ensure optimal BMI before pregnancy and optimal weight gain during pregnancy”. Other reasons for why we focused on pre-pregnancy BMI included 1) the discovery study cohort is of moderate sample size, which limits our ability to explore other less prevalent environmental exposures; 2) compared to other environmental factors, pre-pregnancy BMI is a common variable that is available across discovery and replication samples.

Our findings indicate the potential importance of gene-environment interactions in PTB. More work remains to be done in future studies to explore interactions between genetic variants and other environmental factors. We have added the following comment to the Discussion. “These findings highlight the importance of taking non-genetic factors into account when conducting genetic association studies of PTB. If further validated, these findings may provide new insight into the etiology of PTB and improve our ability to predict and prevent PTB. More work remains to be done in future studies to explore interactions between genetic variants and other environmental factors. ” **[Page 15, Paragraph 2 of the marked manuscript]**

Point-by-point response to Reviewer #1

1. “Reviewer #1 (Remarks to the Author): I think that my comments are well addressed.”

Response: Thank you.

Point-by-point response to Reviewer #2

1.” The authors have done a thorough job, responding to the reviewers' comments. My major reservation remains, however, about the sample size and power for replication, especially considering that this is searching for G*E and not individual SNP effects. On balance, the study is not sufficiently powered to yield a strong conclusion. The reader would be much more confident in the result if the authors were to able to include additional replication data.”

Response: We appreciate this concern and have performed additional analyses to address it. Unlike for conventional GWAS, there is no generally acceptable genome-wide significance threshold for gene-by-environment interactions. For this reason, we adapted an alpha level of 5×10^{-8} as the genome-wide significance threshold, a threshold that has been used by other recent genome-wide G×E interaction studies [PMID: 27723779; 25781442] which assume about 1 million independent tests across the genome ($0.05/1,000,000 = 5 \times 10^{-8}$).

In this revision, we further used a permutation approach to determine the empirical p-value for the identified significant interaction findings, according to suggestions by Reviewer 4. Please refer to our response to Comment #3 from Reviewer 4 for detailed information. Briefly, we permuted the PTB case-control status for 1.9×10^9 times in our discovery sample to ensure a stable estimation of empirical p-value. In this study, the empirical p-value for the rs11161721-pre-pregnancy BMI category interaction is 1.2×10^{-8} , which is similar to our observed p-value ($P_{\text{obs}} = 1.8 \times 10^{-8}$) in the discovery sample. The permutation result lends further evidence that our findings are highly unlikely to be due to chance alone.

We agree that it would be ideal if we could perform another independent replication. Unfortunately, unlike for other phenotypes (e.g., BMI, diabetes), there is only a very limited number of GWAS studies on PTB. We hope that our interesting findings will stimulate more such research in the future.

In the Discussion, we further emphasized the need for additional replications of our findings as follows: “4) we still could not rule out the possibility that our study finding in African Americans is a false positive result, although such chance is extremely low, given all of the analyses and an independent replication that we performed. Further replications in larger African American and Caucasian cohorts are still needed.” [Paragraph 2, Page 13 of the marked manuscript]

Point-by-point response to Reviewer #4

1. “This is an interesting paper investigating a relatively less studied genetic component of preterm birth. As the authors pointed out, studies collecting such data are few and relatively small. The manuscript is much improved with the reviewers comments, and worth pointing out that the replication of GxE finding in the same ethnic group does strengthen the discovery.”

Response: We thank the reviewer for the insightful comments.

2. “- Genome-wide GxE has been difficult due to the heterogeneity in environmental exposures, and lack of replication across studies. In this study, the issue is compounded with small sample sizes. Several methods have been used for assessing the significance of GxE interactions: conventional Bonferroni threshold, permutation, family-wise discovery rate (though the Bonferroni p-value threshold is still by far most commonly used). Suggest the authors permute the case control labels for PTB to get an empirical p-value for the discovery study on top of the meta-analysis with the replication study. See recent paper in PLoS Genetics 27723779.”

Response: The suggestion is well taken. In the revision, we performed the permutation analyses in our discovery sample as suggested. Assuming P_{obs} is the p-value we observed for the significant finding in our discovery sample (in our study, $P_{\text{obs}}=1.8\times 10^{-8}$ for the interaction between rs11161721 and pre-pregnancy BMI category), we defined the minimum number of permutations needed as $1/P_{\text{obs}}$. Since a higher number of permutations may provide a more stable estimation of the empirical p-value, we permuted the PTB case-control status 1.9×10^9 times (more than $30 \times 1/P_{\text{obs}}$ times) in the discovery sample, to ensure a stable empirical p-value. We calculated the p-value for the rs11161721-pre-pregnancy BMI category interaction term in each permuted sample. The empirical p-value for the rs11161721-prepregnancy BMI category interaction was then calculated as the proportion of the permutation realization with a p-value for the interaction term $< P_{\text{obs}}$. In this study, the empirical p-value for interaction is 1.2×10^{-8} , which is similar to our observed p-value ($P_{\text{obs}} = 1.8\times 10^{-8}$) in the discovery sample. When pre-pregnancy BMI was classified as normal weight vs overweight/obesity, then the empirical p-value for the interaction between rs11161721 and overweight/obesity was 5.0×10^{-8} . This permutation result lends further evidence that our findings are highly unlikely to be due to chance alone. We have added the empirical p-value for the interaction association into the revised manuscript.

“Our G×E analyses with the conventional 1-degree of freedom (df) test revealed a genome-wide significant interaction between maternal *rs11161721* (an intronic SNP in the *collagen, type XXIV alpha 1* [*COL24A1*] gene) at 1p22 and pre-pregnancy BMI category on overall PTB risk ($P_{\text{G}\times\text{E}} = 1.8\times 10^{-8}$, empirical $P_{\text{G}\times\text{E}} = 1.2\times 10^{-8}$, **Fig 1a**).” [Paragraph 3, Page 6 of the marked manuscript]

“For the identified significant interactions, we used the permutation approach to determine the empirical p-value in the discovery sample. Assuming P_{obs} is the p-value we observed for the significant interaction finding in our discovery sample, we defined the minimum number of permutations needed as $1/P_{\text{obs}}$.⁵¹ A higher number of permutations will provide a more stable estimation of the empirical p-value. Therefore, we permuted the PTB case-control status 1.9×10^9 times for a $P_{\text{obs}}=1.8\times 10^{-8}$, which is more than $30 \times 1/$

P_{obs} to ensure a stable empirical p-value. We calculated the p-value for the interaction term in each permuted sample. The empirical p-value for the interaction was then calculated as the proportion of the permutation realizations with a p-value for the interaction term $< P_{obs}$.” [Paragraph 2, Page 20]

3. “- I also have some difficulty with the two revised tables: Table 2 and 3. They are a little complicated, and does not allow for quick comparison across the different strata compared to the reference group. Figure 2 is nicely done, and allows for immediate visual assessment of the interaction and differences in the BMI categories across genotypes. Since genotypes are fitted on additive scale, the tables can be simplified to present the main effects of each A allele (for Table 3), overweight/obese and interaction. Suggest to see recent paper in PLoS Genetics 27723779 for simplified table representation. ”

Response: Again, these points are well-taken. In the revision, we have revised the Tables accordingly to simplify the Results section. To allow for a quick comparison on PTB risk across different genetic strata, we simplified Table 2 to present the effects of pre-pregnancy overweight and pre-pregnancy obesity on PTB risk (using normal weight mothers as the reference group), stratified by genotype rs11161721. This table format is similar to Table 1 in the recent published paper (PMID 27723779) as cited by the reviewer. We have also revised Table 3 accordingly to present the main effects of the *rs11161721*-A allele, overweight/obesity and their interaction in both the discovery and the replication samples. Both revised tables are also shown below,

Table 2 (revised). Stratified analyses^a by genotypes of rs11161721 for the association between pre-pregnancy BMI category and PTB in the mothers from the Boston Birth Cohort

Genotype	Normal weight mothers			Overweight mothers				Obese mothers				Pint ^c
	PTB (n)	TB (n)	OR	PTB (n)	TB (n)	OR (95%CI) ^b	P	PTB (n)	TB (n)	OR (95%CI) ^b	P	
rs11161721												
CC	136	293	1.0	140	168	1.8 (1.3-2.4)	4.3×10 ⁻⁴	117	127	2.0 (1.4-2.8)	5.4×10 ⁻⁵	
CA	107	137	1.0	62	78	0.9 (0.6-1.4)	0.60	41	93	0.6 (0.4-0.9)	0.02	
AA	25	17	1.0	9	17	0.4 (0.1-1.2)	0.10	4	15	0.2 (0.0-0.7)	0.02	1.8×10 ⁻⁸

PTB: preterm birth; TB: Term birth; OR: odds ratio; CI: confidence interval: normal weight: pre-pregnancy BMI: 18.5 – 24.9 kg m⁻²; overweight: pre-pregnancy BMI: 25.0 – 29.9 kg m⁻²; obesity: pre-pregnancy BMI ≥30 kg m⁻².

^a Normal weight mothers as the reference group. ^b Adjusted for genotyping batch, genetic ancestry, maternal age at delivery, parity, and infant's gender.

^c The interaction effect was analyzed in the total sample by adding pre-pregnancy BMI category, *rs11161721* (under the additive genetic model) and their interaction term into the regression model, with adjustment of the same covariates as mentioned above.

Table 3 (revised). The main effects of maternal *rs11161721*-A allele, pre-pregnancy overweight/obesity and their interaction effects on PTB in the mothers from the Boston Birth Cohort and from the GPN study

Variable	BBC discovery ^a n=1586			GPN Replication ^b African-American Mothers n=300			GPN Replication ^b Caucasian Mothers N=683		
	OR	95% CI	P	OR	95% CI	P	OR	95% CI	P
	rs11161721 - A allele	1.8	1.4-2.3	9.8×10 ⁻⁶	1.4	0.7-2.4	0.32	0.9	0.7-1.2
OWO	1.9	1.5-2.5	2.6×10 ⁻⁶	1.9	1.0-3.4	0.04	1.2	0.8-1.9	0.36
rs11161721×OWO interaction	0.4	0.3-0.5	5.9×10 ^{-8c}	0.3	0.2-0.8	0.01	1.2	0.7-2.0	0.39

BBC: Boston Birth Cohort. PTB: preterm birth; OWO: overweight or obesity (pre-pregnancy BMI ≥25 kg m⁻²), with normal weight mothers as the reference group; OR: odds ratio. CI: confidence interval. GPN: Genomic and Proteomic Network for Preterm Birth Research.

^a The analysis was conducted using the logistic regression model, adjusted for genotyping batch, genetic ancestry, maternal age at delivery, parity, and infant's gender in the BBC discovery sample. ^b The analysis was conducted using the logistic regression model, adjusted for genetic ancestry, parity and infant's gender in the GPN replication sample. ^c Here *P* value for the interaction effect was estimated using an interaction term of pre-pregnancy OWO (coded as 0=NW, 1=OWO) and the rs11161721 genotype (additive genetic model), rather than an interaction term of the pre-pregnancy BMI category (coded as 0=NW, 1=OW, 2=OB) and rs11161721 (additive genetic model) as shown in Table 2.

4. “- I agree with reviewer one that this finding is still immature for personalised prevention, especially with the power calculations presented for the Caucasians. ”

Response: We agree completely. In the revision, we have toned down our conclusion in the revised Discussion, and emphasized that our findings need further validation in other independent cohorts.

“Such findings may reveal etiologic genes/pathways interacting with environmental factors, and identify possible drug targets. Our findings, if further validated in other independent cohorts, could also prove valuable for the prediction and prevention of PTB in African Americans, since pre-pregnancy BMI is a modifiable factor and both *rs11161721* genotypes and pre-pregnancy BMI can be easily measured well before pregnancy.” [Paragraph 2, Page 12 of the marked manuscript]

5. “- Results section can be more concise. The section on sensitivity of the variant rs11161721, the first paragraph may be shorten to only refer to Figure 2. The OR comparison in stratified genotype classes in Table 2 could be moved to Supplementary. It may also be more accurate to say "consistency of interaction across PTB subtypes"”

Response: We have shortened our Result section accordingly, and done our best to convey our findings more clearly [Paragraph 2, Page 7 of the marked manuscript]. The OR comparison in stratified genotype classes for PTB subtypes, which was part of the original Table 2, has been moved to Supplementary Table 4. The title for Supplementary Table 4 has been changed to “Consistency of the *rs11161721* × pre-pregnancy BMI interaction effect across PTB subtypes in 1,586 African American mothers from the Boston Birth Cohort”, and the sub-title for the related text in the Result section has been changed to “*rs11161721* × pre-pregnancy BMI interaction, and its consistency across PTB subtypes”.

Point-to-point response

Manuscript No # NCOMMS-16-07795

Title: Genome-wide approach identified a novel gene-maternal pre-pregnancy BMI interaction
on preterm birth

Point-by-point response to comments from the reviewers

1. REVIEWERS' COMMENTS:

Reviewer #4 (Remarks to the Author):

“It is reassuring that the same conclusions are reached with permutations. Tables have been simplified and made more readable. Note font size of Table 2 footnotes. As with many genome-wide GxE and replications, authors have made clear the limitations, including less strong conclusions on the utility of current findings in personalised medicine. The authors have addressed all my concerns adequately.”

Response: Thank you. We have updated the font size of the Table 2 footnotes accordingly.